# A topological refactoring design strategy yields highly stable granulopoietic proteins

Julia Skokowa [1,8✉], Birte Hernandez Alvarez [2,8], Murray Coles [2], Malte Ritter[1], Masoud Nasri[1], Jérémy Haaf[1], Narges Aghaallaei[1], Yun Xu[1], Perihan Mir[1], Ann-Christin Krahl[1], Katherine W. Rogers [3,5], Kateryna Maksymenko[2,3], Baubak Bajoghli [1], Karl Welte[1], Andrei N. Lupas [2], Patrick Müller [3,6] & Mohammad ElGamacy [1,3,4,7✉]

Protein therapeutics frequently face major challenges, including complicated production, instability, poor solubility, and aggregation. De novo protein design can readily address these challenges. Here, we demonstrate the utility of a topological refactoring strategy to design novel granulopoietic proteins starting from the granulocyte-colony stimulating factor (G-CSF) structure. We change a protein fold by rearranging the sequence and optimising it towards the new fold. Testing four designs, we obtain two that possess nanomolar activity, the most active of which is highly thermostable and protease-resistant, and matches its designed structure to atomic accuracy. While the designs possess starkly different sequence and structure from the native G-CSF, they show specific activity in differentiating primary human haematopoietic stem cells into mature neutrophils. The designs also show significant and specific activity in vivo. Our topological refactoring approach is largely independent of sequence or structural context, and is therefore applicable to a wide range of protein targets.

[1] Division of Translational Oncology, Department of Hematology, Oncology, Clinical Immunology and Rheumatology, University Hospital Tübingen, 72076 Tübingen, Germany. [2] Max Planck Institute for Biology, 72076 Tübingen, Germany. [3] Friedrich Miescher Laboratory of the Max Planck Society, 72076 Tübingen, Germany. [4] Heliopolis Biotechnology Ltd, Cambridge CB24 9RX, UK. [5] Present address: Division of Developmental Biology, Eunice Kennedy Shriver National Institute of Child Health and Human Development, National Institutes of Health, Bethesda, MD 20892, USA. [6] Present address: Department of Biology, University of Konstanz, 78464 Konstanz, Germany. [7] Present address: Max Planck Institute for Biology, 72076 Tübingen, Germany. [8] These authors contributed equally: Julia Skokowa, Birte Hernandez Alvarez. ✉email: julia.skokowa@med.uni-tuebingen.de; mohammad.elgamacy@tuebingen.mpg.de

D e novo protein design can serve as a powerful tool for protein therapeutics discovery and development, having enabled unprecedented strides in navigating the protein sequence and structure spaces[1,2], and tailoring novel functions[3]. A central objective in the design of functional proteins is to tailor a scaffold structure that best supports the function encoded on a fraction of the template protein's surface. In addition to high activity levels, other criteria are pivotal for the successful deployment of a protein drug, such as activity half-life, folding rates, structural stability, solubility, and molecular weight[4,5]. These properties depend on a set of controllable protein parameters such as a protein's sequence, topology, and size.

Computational design can offer control over these parameters to radically alter the final properties of a protein. The design process can be carried out under the structural constraints encoded by the prior knowledge of the active epitope. First, the sequence can be optimised to maximise the enthalpies of the core residues ("core repacking") or to minimise the solvent-exposed hydrophobic residues, with the aim to improve the thermostability and the solubility, respectively. Second, topology can be changed to objectively optimise folding rate and folding free energy. The topological contact order, a metric that describes the average sequence distance of contacting residues, has been shown to inversely correlate with the logarithm of folding rates[6]. Topological simplification also tends to affect the folding thermodynamics, for instance, removing long, flexible loops tends to decrease the absolute difference in entropy between the unfolded and folded states (i.e. $|\Delta S|$)[7,8]. Third, a protein's size has significant influence on its folding rate[6], as well as on its folding free energy, where the latter is impacted by the magnitude of configurational entropy change upon folding, that, in turn, increases quadratically with the length of the polypeptide chain[9].

In this work, we demonstrate a generalisable refactoring strategy, which aims at preserving a protein's functional epitope, while reconstructing the protein's fold and composition around it. The basis of the proposed strategy relies on constructing more than one structured loop de novo into the same protein at once, which enables the reordering of secondary structural elements along the primary structure. This simplifies the topology from around the active epitope, disposes of entire secondary structures along the sequence, and reduces the protein size down to the necessary scaffolding required to maintain the structural integrity of the functional epitope. Furthermore, spatially distant residues from the functional epitope are optimised to best encode information necessary to improve folding or solvation free energies.

Here, we sought to demonstrate this in silico design strategy as a means for conceiving novel receptor modulators by remodelling a natural cytokine: the granulocyte-colony stimulating factor (G-CSF) (Fig. 1A). G-CSF is a cytokine that stimulates the proliferation and myeloid differentiation of haematopoietic stem and progenitor cells (HSPCs) in the bone marrow and mobilises them into the blood stream[10]. Recombinant human G-CSF (rhG-CSF) has demonstrated great immunotherapeutic utility due to its potent activity in the stimulation of granulopoiesis, boosting the immunity of neutropenic patients, who suffer from a severely reduced number of neutrophils due to genetic factors or chemotherapy[11–16]. Like most other protein therapeutics, rhG-CSF is clinically deployed in its native form or with a few modifications. This is reflected in several suboptimal pharmaceutical features, such as its low recombinant production yield, poor solubility and stability, and short shelf- and serum half-lives. Only classical engineering strategies have thus far been pursued to improve rhG-CSF, spanning point-mutagenesis[17–20], PEGylation[21,22] and circularisation[23,24].

Our strategy aimed to minimise the loss of entropy during folding and receptor binding by reducing structural complexity and rigidifying the bound conformation, respectively (Fig. 1). This resulted in designs that possess minimal sequence similarity with the native G-CSF and contain no disulfide bonds, representing a miniaturised G-CSFR binding domain. We evaluated the biophysical properties of the resulting proteins and determined the structure of the most active one, which showed atomic-level agreement with the design. We found that the designed proteins, especially in tandem, are highly effective receptor activators, and are potent and specific in inducing proliferation and differentiation of primary human haematopoietic stem cells into functional neutrophils. Strikingly, the designs also had significant granulopoietic activity in vivo in zebrafish and mice.

## Results

**Computational design of G-CSFR binders.** Human G-CSF induces JAK/STAT signalling and downstream granulopoiesis by binding to the ectodomains of the G-CSF receptor (G-CSFR)[25]. Biochemical and structural studies have shown that binding site II—comprising residues K16, E19, Q20, R22, K23, D27, D109, and D112—is the dominant binding motif on the surface of G-CSF (Fig. 1A). Our refactoring process sought to reduce the topological complexity, improve thermodynamic and kinetic stabilities, reduce the size, increase the solubility, and eliminate the need for post-translational modifications (PTMs) in G-CSF. The PTMs in human G-CSF are comprised of two disulfide bridges across C69 and C75 and C97 and C107, and an O-glycosylation at T133 (Swiss-Prot numbering). The refactoring process was achieved by rewiring the topology (Fig. 1B), repacking the core, and optimising solvent-exposed residues (Fig. 1C). The first step was to eliminate two segments from the primary structure that are 32- and 22-residue long, which structurally encode three long loops and a helix. This reduced size architecture was redesigned to optimise the packing of core residues, with the goal of rigidifying the binding epitope. The initial structural template was extracted from the G-CSF–G-CSFR complex structure (PDB: 2D9Q)[26], which shows that G-CSF has a similar binding conformation when compared to unbound G-CSF structures. Moreover, solvent-exposed residues distal to the binding epitope were also designed to replace hydrophobic side chains that were used to efface the removed motifs and to enhance the helical character of the assembled protomers (see the "Methods" section). While the design protocol used optimised decoys for their *talaris13* energy score, after several rounds of design the final hits were filtered purely based on their packing quality. The top several hundred candidates were subject to equilibrium molecular dynamics (MD) for the inspection of their structural stability, where the most structurally stable decoy was forwarded for the next design stage (loop design). At the end of this first step it was critical to choose a single decoy for the following loop design, as the design protocol entails unrestrained backbone motions that would generate diverse conformations at the disjoint junctions.

The second step aimed at arriving at single-chain variants to minimise the contact order, which was done through the design of two short loops across the gaps. The two gaps extended approximately across 10 and 13 Å, hence we sought to connect them by 3- and 4-residue de novo loops, respectively. Since the sequence space of 3- or 4-residue loops is sufficiently tractable for nearly exhaustive exploration, we picked the amino acids with the highest loop propensity and generated all the possible sequence combinations. For the 3-residue loop, 9 amino acid types were used to generate 729 ($9^3$) combinations, and for the 4-residue loop 8 amino acid types were used to generate 4096 ($8^4$) combinations (see the "Methods" section). These sequences were modelled in and refined across their respective junction. This was followed by conformational homogeneity evaluation across

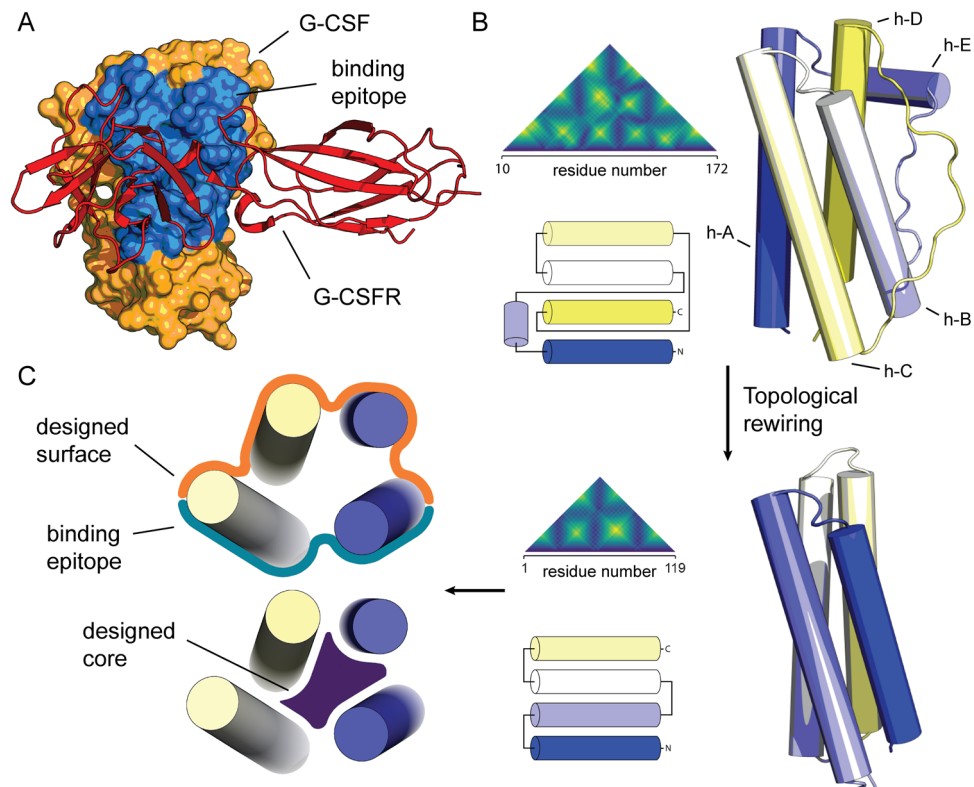

**Fig. 1 Computational design stages of the topological refactoring strategy. A** X-ray structure of G-CSF (orange) bound to its cognate receptor (red) through its binding epitope (blue). **B** According to the *topological refactoring* strategy, the topology of the native G-CSF was rewired from around the fixed binding epitope. This was done by replacing the two long, bundle-spanning loops, by two short, de novo designed loops; reconstructing an up–up–down–down bundle into an up-down helical bundle. This resulted in much simpler contact maps, a lower contact-order, and a smaller protein (G-CSF residue numbering is based on PDB: 2D9Q). **C** The design models were further optimised in order to idealise the core packing (blue volume) and to hydrophilise residues distal from the binding epitope (orange crust).

repeated MD simulations. Initially, loops across each junction were modelled individually, whereby the other loop was modelled as a constrained tri- or tetra-glycine. The 50 conformationally most stable loops were from each junction, and combined together to generate 2500 spliced models ($50^2$). The simulations were repeated for the combined loops models and the four most conformationally stable molecules were chosen for experimental evaluation.

The four designs were named Boskar1–4, and were 123-amino acid long—a substantial size reduction from rhG-CSF, which is 174 amino acid-long. Moreover, the reshuffling of the secondary structural elements along the primary structure during the topological reconstruction, combined with the sequence optimisation, resulted in low full-length sequence homology, where the sequence identity with rhG-CSF is below 32% for all of the designs (Supplementary Table 1). In addition to their new topology, lower contact order, and idealised sequences, the designs were also devoid of the native form's PTMs. The designs lacked glycosylation sites and disulfide bonds, which are normally important for hG-CSF folding. In fact, our proteins were designed not to contain any cysteine residues, which have been implicated in misfolding of rhG-CSF[17].

**The designs possess granulocytic proliferative potential**. To query the potential biological activity of the four experimentally tested designs, we used murine NFS-60 cells, a hematopoietic cell line that is routinely deployed to quantify G-CSF-triggered proliferative potential[27]. Cell densities were assessed by a fluorescence redox-based assay 48 h after treatment with the designs or

rhG-CSF, where the average $EC_{50}$ values (i.e. the concentration that gave half-maximal cell density response) ranged from micromolar to nanomolar (Supplementary Table 1). Boskar1, Boskar2, Boskar3, and Boskar4 showed $EC_{50}$ values of $164.5 \pm 19.9$, $241.8 \pm 74.3$, $58.5 \pm 6.0$ and $2.05 \pm 0.2$ nM, respectively. The most active designs, Boskar3 and Boskar4, exhibited dose–response curves in the nanomolar range upon 48-hour treatment of NFS-60 cells (Fig. 2A). We then set out to investigate the dose- and time-response kinetics of the two most active designs, which showed that our designs possess slower proliferation induction kinetics than rhG-CSF. The proliferative concentrations Boskar4 and Boskar3 were found to reach sub-nanomolar levels for longer-duration treatments in time-lapse microscopy analyses of the cell proliferation kinetics over longer treatment durations (Fig. 2B–D, Supplementary Fig. 1 and Supplementary Movies 1–4).

**Activation of G-CSFR signalling by Boskar3 and Boskar4**. To evaluate the dependency of the response to the designed proteins on G-CSFR expression, we knocked out G-CSFR in NFS-60 cells using CRISPR/Cas9-mediated mutagenesis. For this, we synthesised guide RNA (gRNA) specifically targeting exon 4 of *CSF3R* (cut site: chr4 [+126,029,810:−126,029,810]) to introduce stop-codon or frameshift mutations in the extracellular part of all G-CSFR isoforms. We generated pure G-CSFR KO NFS-60 cell clones that have one nucleotide deletion on each allele, as assessed by Sanger sequencing and tracking of indels by decomposition (TIDE) analysis (Supplementary Fig. 2)[28]. In contrast to wild type cells, G-CSFR KO NFS-60 cells did not respond to treatment with

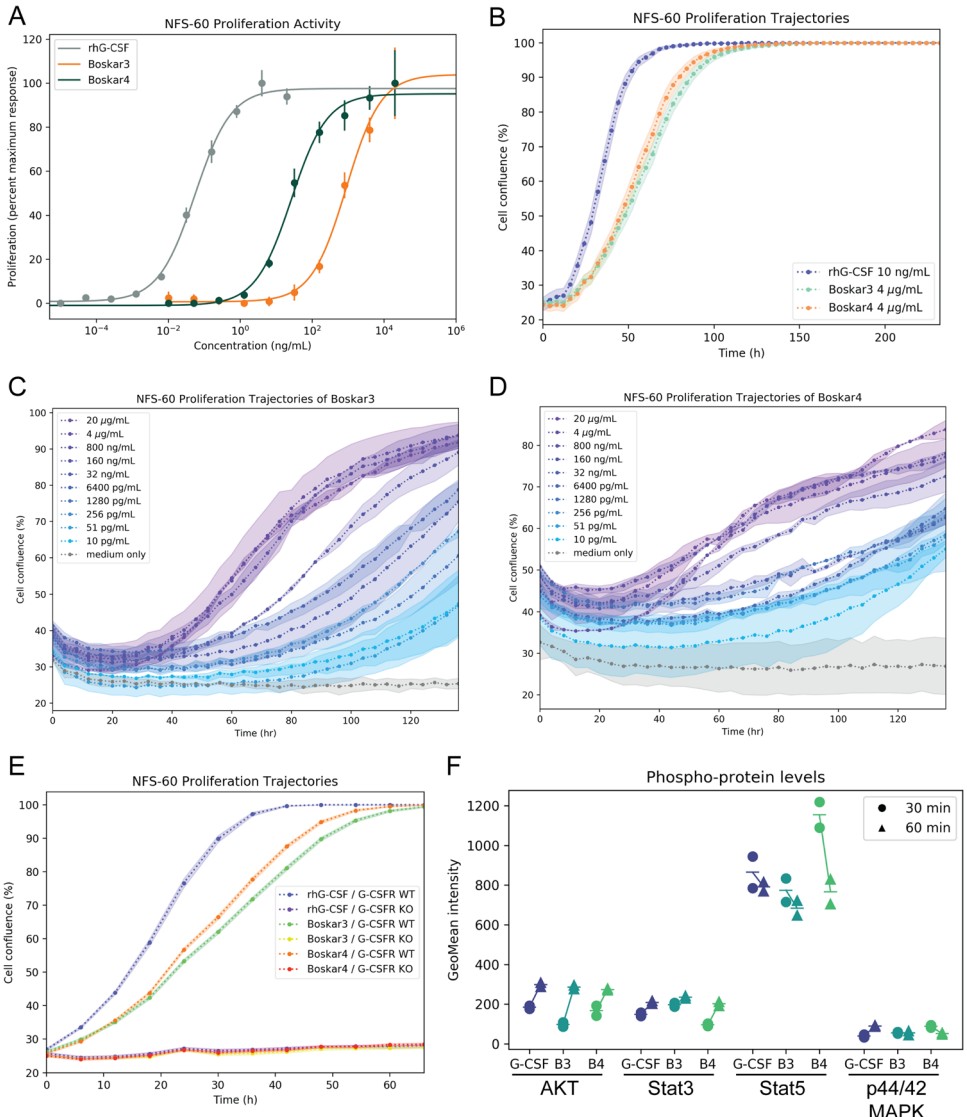

**Fig. 2 The designs display potent and specific activity in cell-based assays. A** Dose–response curves of NFS-60 proliferation upon 48 h treatment with Filgrastim (rhG-GCSF), Boskar3, or Boskar4. Datapoints and error bars represent mean ± standard deviations of three biologically independent replicates. **B** Time-dependent proliferation trajectory of surface-immobilised NFS-60 cells over a 10-day treatment. Data points and shades represent mean and standard deviation values of three biologically independent replicates. Experiments were performed three times in triplicates. Data of one representative experiment is shown. **C, D** Dose- and time-dependent proliferation trajectories over a 5-day treatment of free-floating NFS-60 cells, under the influence of Boskar3 and Boskar4 treatments, respectively. Data points and shades represent mean and standard deviation values of three biologically independent replicates. **E** G-CSFR-deficient primary stem cells (G-CSFR KO), show abolished proliferative responses to either rhG-CSF or the designs. Experiment was performed twice in triplicates. Data points and shades represent mean and standard deviation values of three biologically independent replicates. **F** Intracellular levels of phospho-AKT (Thr308), phospho-ERK1/2 (p44/42 MAPK), phospho-STAT3 (Tyr705), and phospho-STAT5 (Tyr694) in CD34$^+$ HSPCs treated with rhG-CSF or the designs (see the "Methods" section). Geometric mean of the expression intensity of each phospho-protein (GeoMean intensity) is shown on the y-axis. The experiment was performed twice.

rhG-CSF, Boskar3 or Boskar4 (Fig. 2E). These data demonstrate that the designed proteins act via G-CSFR.

Binding of G-CSF to G-CSFR rapidly activates a cascade of intracellular events, including phosphorylation of downstream effectors, e.g. Akt, STAT3, STAT5 or MAPK, that ultimately induce granulocytic differentiation[29]. To test whether our designed proteins directly induce G-CSFR signalling, we measured these immediate phosphorylation targets of G-CSFR signalling in CD34$^+$ HSPCs. Indeed, we found that Akt, STAT3, STAT5 and p44/42 MAPK (Erk1/2) were tyrosine phosphorylated in HSPCs treated with Boskar3 or Boskar4 to a similar degree as in rhG-CSF-treated cells (Fig. 2F). Together, this shows that the biological activity of the designs is directly attributable to G-CSFR activation.

**The designed protein Boskar4 has enhanced stability characteristics.** Expression of the designs in *E. coli* showed that all designs were highly expressed in the soluble cell fraction. We focused our biophysical and structural characterisation on one design; Boskar4, which showed the highest activity in endpoint dose–response assays. In contrast to rhG-CSF which is expressed insolubly to a yield of 3 mg/l culture and has to be refolded[30], Boskar4 was expressed solubly in *E. coli* to a yield of >80 mg/l culture as final yield after tandem affinity and size-exclusion chromatography. The solubility of Boskar4 also outstripped that of rhG-CSF, where Boskar4 could be readily concentrated to >20 mg/ml in PBS buffer, while rhG-CSF precipitated above a concentration of about 4 mg/ml in PBS buffer. Circular dichroism

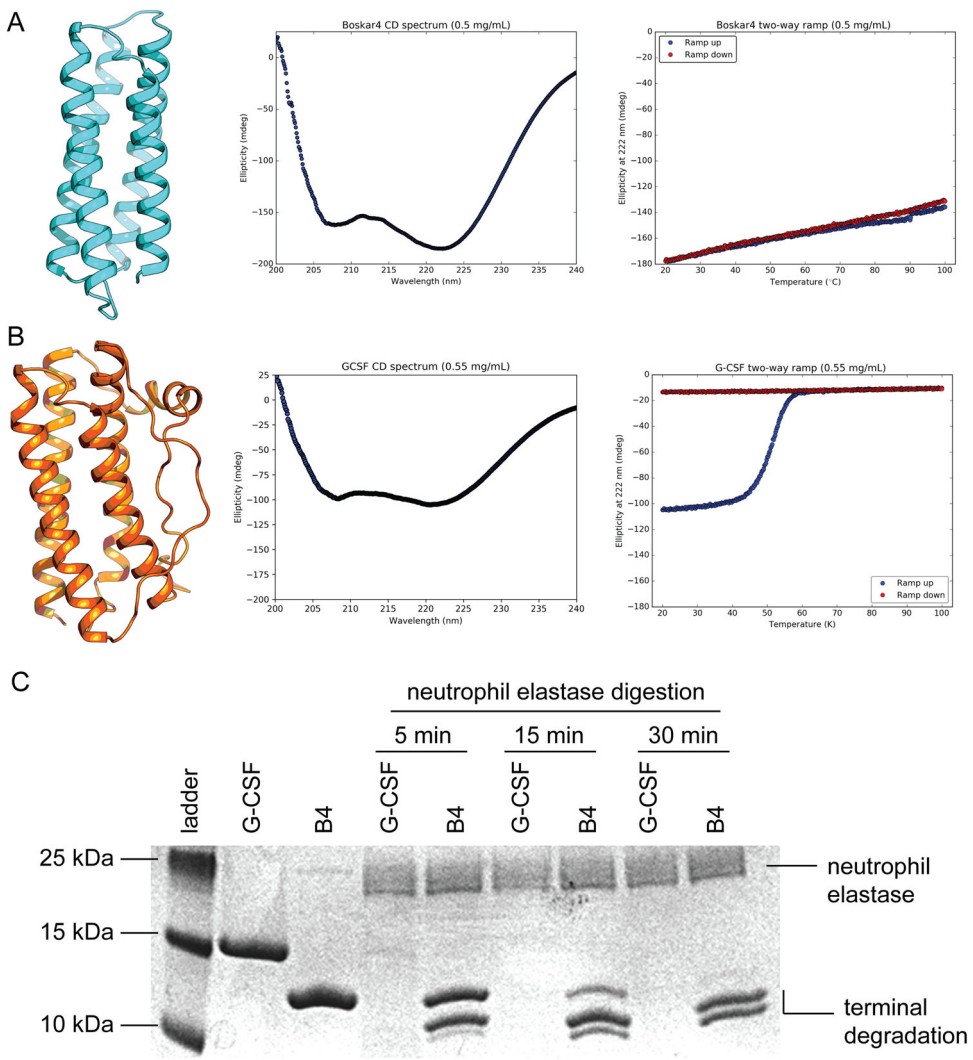

**Fig. 3 The designed protein Boskar4 is substantially more stable than rhG-CSF. A** Design model of Boskar4 (left pane) has a CD spectrum that shows a strong helical content (middle pane). The melting curve of Boskar4 shows exceptional stability since it does not undergo a melting transition up to 100 °C (blue curve) and shows a full regain of residual ellipticity at 222 nm upon cooling (red curve). **B** In contrast, rhG-CSF appears to display weaker helical signal compared to the design (middle pane), and melts irreversibly at 57 °C. **C** Neutrophil elastase treatment of Boskar4 (B4) and rhG-CSF shows Boskar4 to be substantially more resistant to proteolytic digestion in comparison to rhG-CSF. Expected neutrophil elastase and degradation bands are indicated.

(CD) spectra of Boskar4 show it to possess strong helical character, with almost two-fold the mean residual ellipticity magnitude of rhG-CSF, when measured at the same concentration (middle pane of Fig. 3A, B). CD spectroscopy was also used to assess thermal unfolding of Boskar4 and rhG-CSF. Boskar4 appeared to remain stably folded up until 100 °C, with completely reversible residual ellipticity recovery upon cooling. Conversely, the melting curves of native rhG-CSF showed irreversible thermal unfolding at a temperature of 57 °C, whereby the protein heavily precipitates upon cooling (right pane of Fig. 3A, B). To investigate if the disulfide bond formation has an impact on the irreversible denaturation or thermostability of rhG-CSF, we sought to repeat these melting and cooling experiments with addition of reducing agents to the buffer. Due to high buffer absorbance, we carried out these experiments using nanoDSF, for rhG-CSF in standard PBS buffer, PBS buffer supplemented with a mixture reduced and oxidised glutathione (2.5 mM and 0.5 mM, respectively), or in PBS buffer supplemented with 5 mM dithiothreitol (DTT). These melting curves showed similar melting points in either PBS alone or in a mixture of reduced and oxidised gluthathione, as the latter condition should accelerate the equilibrium for formation or

breaking of disulfide bonds, at 50 and 51.6 °C, respectively (Supplementary Figs. 3 and 4). Conversely, the purely reducing effect of DTT resulted in a lower thermostability of rhG-CSF where the melting point was 35.4 °C, and a clear scattering change could be observed, indicative of aggregation (Supplementary Fig. 5). In all three cases, the unfolding of rhG-CSF was irreversible.

To test whether the enhanced thermostability of the designs can be of potential pharmacokinetic advantage, we tested the protease resistance of Boskar4 in comparison to rhG-CSF. Neutrophil elastase (NE) is a serine protease that has broad substrate specificity. Secreted by neutrophils and macrophages during inflammation, NE enzymatically degrades G-CSF, inhibiting its activity, which constitutes a negative feedback loop against excessive cytokine-induced granulopoiesis[31,32]. Analysing the products of NE treatment showed that rhG-CSF was almost entirely digested after a five-minute-treatment (Fig. 3C). Boskar4, however, was largely protease-resistant even after 30 min. Most of the partial digestion of Boskar4 appears to be restricted to terminal fragments, possibly representing the unstructured hexa-Histidine and thrombin cleavage purification tag. Therefore, we

sought to identify the composition of these degradation bands using nanoflow liquid chromatography–tandem mass spectrometry (nano-LC–MS/MS). While the mass spectrometry analysis (Supplementary Fig. 6) for the band corresponding to full-length Boskar4 shows complete sequence coverage, peptides of partial degradation bands 1 and 2 do not cover, or only partially cover, the N-terminus and C-terminus, indicating these bands to be terminal degradation products of Boskar4 by neutrophil elastase.

To expand our analysis, we further tested the protease sensitivity of Boskar4 and rhG-CSF against a set of diverse human proteases representing different classes of proteolytic enzymes. These were the lysosomal aspartyl protease Cathepsin D, the metalloprotease ADAM10, the cysteine protease Cathepsin L, and the serine protease Cathepsin G. While these results showed both Boskar4 and rhG-CSF are not sensitive to Cathepsin L or ADAM10, rhG-CSF was sensitive to Cathepsin D and Cathepsin G degradation, whereas Boskar4 was resistant to them under the same conditions (Supplementary Fig. 7). Collectively, these results indicate that Boskar4 is less likely to partially or fully unfold, and hence less accessible to proteolytic digestion.

**The solution structure of the Boskar4 design matches its design model to atomic-accuracy**. To evaluate the structural precision of the design process we determined the solution structure of Boskar4, since it showed well-dispersed NMR spectra (Supplementary Fig. 8). The structure was determined using the CoMAND method (Conformational Mapping by Analytical NOESY Decomposition), a protocol that provides unbiased structure determination driven by a residue-wise R-factor tracking the match between experimental and back-calculated NOESY spectra[33]. In the CoMAND protocol, a 3D-CNH-NOESY spectrum[34] is divided into 1D sub-spectra (referred to as strips), each representing contacts to a single backbone amide proton, thus representing the structural environment of the respective residue (see the "Methods" section). Spectral decomposition is then performed to yield local backbone dihedral angles for all residues where strips are available, which are used for initial model building. In subsequent stages, the R-factor is used both to extract further geometric data and as a selection criterion for frame-picking from MD trajectories, yielding the final structure ensemble. For Boskar4, the CNH-NOESY spectra provided 107 strips for CoMAND analysis. Of the 118 non-proline residues, 11 were not observed, presumably due to fast exchange with solvent. Accordingly, all of these residues are at the C-termini of helices or the subsequent loops, including 5 residues (K89-R93) in the α3-α4 loop. We performed CoMAND factorisation calculations for 98 of these strips (strips containing significant overlapped intensities were excluded at this stage), yielding backbone dihedrals that were used as input for model building in Rosetta. The angles obtained were both consistent with the values predicted from chemical shift profiles by TALOS-N[35] (Supplementary Fig. 9). To refine this initial model, we applied a protocol employing alternate rounds of unrestrained and restrained molecular dynamics simulations, with the aim of producing a set of unrestrained structures close to the average structure. To achieve this we applied restraints derived from R-factor optimisation and from conventional "boot-strapping" based on the initial model. These included 76 inter-helical distance restraints (Supplementary Table 3 and Supplementary Fig. 10), primarily between well-resolved aromatic and methyl groups, derived from two further NOESY experiments (see the "Methods" section). We then selected a final ensemble of 17 structures from this pool by R-factor optimisation over all available strips, resulting in an average R-factor of 0.31 ± 0.11 (Fig. 4A, C, Supplementary Table 2). The ensemble deviated by an average of 0.80 Å from the

average structure, and 2.0 Å from the design model (Fig. 4A). Locally aligning the NMR ensemble to the designed binding epitope residues showed a backbone RMSD of 1.0 Å and an all-atom RMSD of 1.2 Å (Fig. 4B), thus demonstrating atomic precision in re-sculpting the binding epitope.

The overall structure of Boskar4 is very similar to the design model in the lengths of the four helices, the bundle architecture, helical register and packing. On the global level, the largest deviation is across narrow end of the bundle (corresponding to the α1–α2 and α3–α4 loops), where the helices of the solution structure are less tightly packed than for the model. This difference is well captured by R-factor optimisation for A97, which shows contacts to methyl groups of L22 and I25 (Supplementary Fig. 11). This region is also the site of the largest deviation on the local level, where the capping motif for the α1 helix of the design is incompatible with the CNH-NOESY data (Supplementary Fig. 12). Here the CoMAND analysis identifies a backbone polymorphism at L22, where 4 of the 17 models adopt an alternate conformation (Supplementary Fig. 12). To take this into account, we rebuilt the α1–α2 loop, constructing a range of trial structures for each of the alternate forms and selected frames to minimise R-factors across the segment. Under the NMR measurements conditions, Boskar4 appeared to be monomeric and thermostable, where temperature and concentration ramps did not influence the reference $^1$H–$^{15}$N HSQC spectrum (Supplementary Fig. 13).

To further examine the internal dynamics of the Boskar4 structure, we measured $^{15}$N {$^1$H}-heteronuclear NOE values and water exchange rates for backbone amide protons. The former report on internal motions over fast (i.e. sub-nanosecond) timescales, while the latter primarily report on hydrogen bond stability on timescales of milliseconds to seconds. Both measures are consistent with stable secondary structure elements; NOE values indicative of high-amplitude fast motions (NOE < 0.8) and shorter exchange times are only observed in the inter-helical loops (Fig. 4D). Thus, the Boskar4 design provides a stable and accurate scaffold for presenting the binding epitope. Analysis of the solubility character of the Boskar4 NMR structure (PDB: 7NY0; 1st model) and G-CSF crystal structure (PDB: 2D9Q), indicates that sequence- and structure-based solubility prediction methods show fewer exposed hydrophobic outliers on the Boskar4 surface, as compared to G-CSF (Supplementary Fig. 14).

**The designs binding to human G-CSFR is enhanced by bivalency**. To characterise the kinetics and affinity of interactions between the designs and the G-CSF receptor, we performed surface plasmon resonance-based measurements for Boskar3, Boskar4 in comparison to rhG-CSF (Fig. 5A–C, Table 1, and Supplementary Fig. 15A–C). Analysis of the kinetics across the injection dilution series, assuming 1:1 binding, resulted in dissociation constants ($K_d$) of 156 and 195 nM for Boskar3 (Fig. 5B), Boskar4 (Fig. 5C), respectively. In comparison, the $K_d$ determined for rhG-CSF was 1 nM (Fig. 5A). Previous studies have reported $K_d$ values for the G-CSF:G-CSFR interaction between 200 pM using SPR[36] and 1.4 nM using ITC[37].

We noticed that titrating higher concentrations of designs (i.e. 10–50 μM) resulted in sensograms that better fit with a 2:2 binding model. Size-exclusion chromatography of the Boskar3 (Supplementary Fig. 16) and Boskar4 (Supplementary Fig. 17) designs indeed showed that the designs partition between monomeric and dimeric forms. This monomer–dimer equilibrium however seems to follow a very slow interchange, even at room temperature (Supplementary Figs. 18 and 19). Nonetheless, in all of the measurements and assays described, both monomer and dimer fractions were pooled together and used as a single

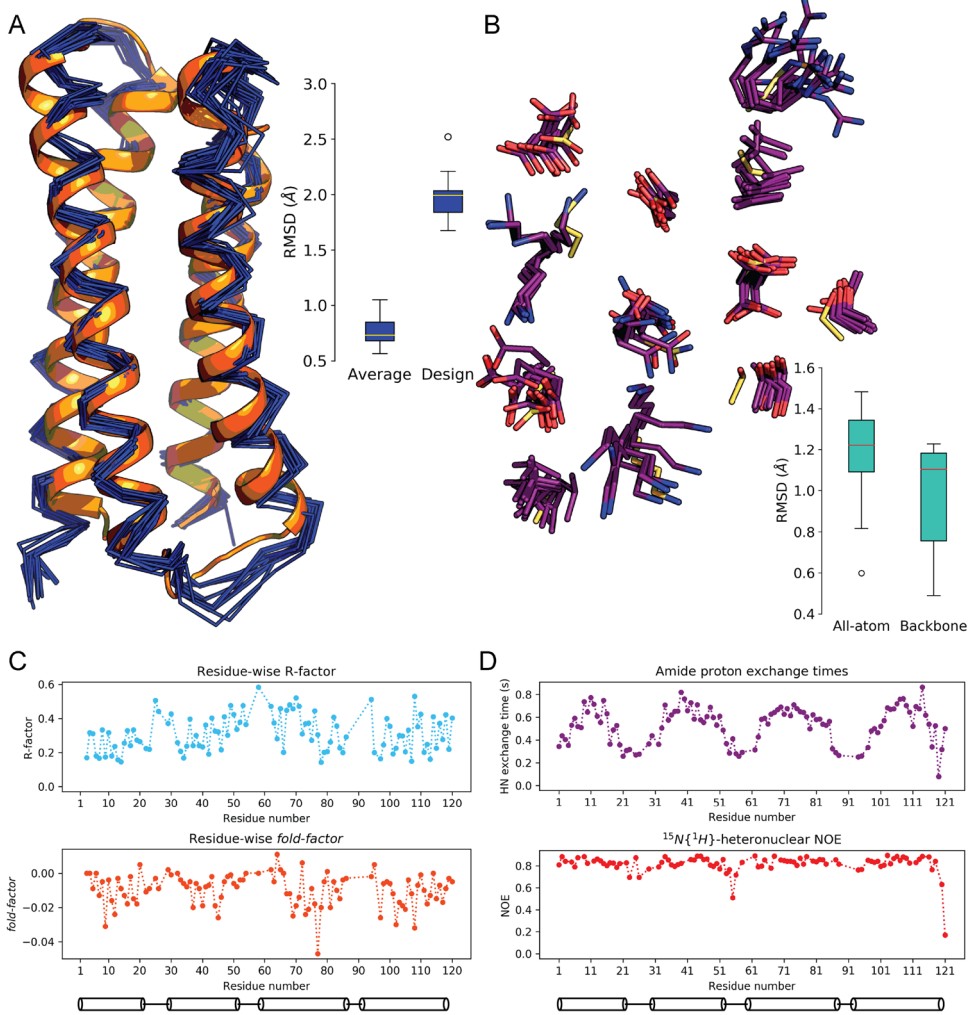

**Fig. 4 The design model shows atomic-level agreement with its NMR solution structure. A** Boskar4 solution structure shows an ensemble deviation from the average structure of 0.8, and 2.0 Å from the designed coordinates. The design model (orange) is shown against the NMR ensemble (dark blue). The boxplot shows the median and spread across the ensemble frames. **B** The backbone atoms RMSD of the binding epitope averaged at 1.0 Å, while all-atom RMSD of averaged at 1.2 Å, highlighting the design precision. The design model residues (yellow) are shown against the NMR ensemble (purple), and the box plot shows the deviations across the ensemble. The boxplot shows the median and spread across the ensemble frames. The box plots represent median (central line), 1st and 3rd interquartile range (box), and minimum and maximum of 1.5 times the latter interquartile range (whiskers). **C** Residue-wise *R*-factor values for the covered residues are shown (top), indicating the normalised CNH-NOESY RMSD from the expected spectra by the final ensemble. The *fold-factor* shown (bottom), a metric which signifies the *R*-factor reduction by non-local residue-residue contacts, shows the contribution of buried helical residues to Boskar4 folding. **D** Both amide proton exchange (top) and $^{15}N\{^1H\}$NOEs (bottom) show the backbone amides of the helical residues to be solvent-shielded and conformationally rigid, respectively.

sample. We next sought to construct a truely bivalent design form by linking two Boskar4 domains in tandem by means of a 24-residue linker (i.e. $(GGGGSS)_4$), named Boskar4_t2 (Fig. 6A). Boskar4_t2 binding to G-CSFR showed a $K_d$ of 5 nM (Fig. 5D, Table 1, Supplementary Fig. 15D), which constitutes a 39-fold improvement of affinity in comparison to Boskar4. This further suggested that Boskar4_t2 may exhibit avidity, thus can be a more efficient dimeriser and subsequently a more active G-CSFR agonist.

**G-CSFR activation is sensitive to its dimerisation spacing.** To study the biological relevance of this affinity enhancement of Boskar4_t2, we evaluated its proliferative potential through end-point NFS-60 proliferation assay, where it showed an $EC_{50}$ of 180 nM; 11-fold better than Boskar4 (Fig. 6A, C, and Supplementary Table 1). We reasoned this improved activation may be

sensitive to the receptor subunit spacing as was reported with other cytokine receptors[38–40]. Therefore, we created another 2-copy tandem construct with a shorter (6-residue) flexible linker (i.e. GGGGSS), named Boskar4_st2. While Boskar4_st2 showed similar affinity to Boskar4_t2 against G-CSFR (i.e. 6 nM; Table 1), it was indeed even more active in inducing proliferation of NFS-60 cells ($EC_{50}$ = 7.6 pM), with a 263-fold improvement in activity over the single domain Boskar4, and 24-fold over the two-domain construct Boskar4_t2 (Fig. 6B, C, and Supplementary Table 1). Repeating the dose- and time-dependent proliferation assays resulted in a similar picture, emphasising the activity gain from efficient dimerisation and linker length reduction (Fig. 6E, F). To further evaluate the generality of this observation, we created the Boskar3_st2 construct, and a mixed Boskar3_Boskar4 construct, where either two Boskar3 domains or one Boskar3 and one Boskar4 domains are connected in tandem using a 6-residue linker (i.e. GGGGSS), respectively. The same pattern was

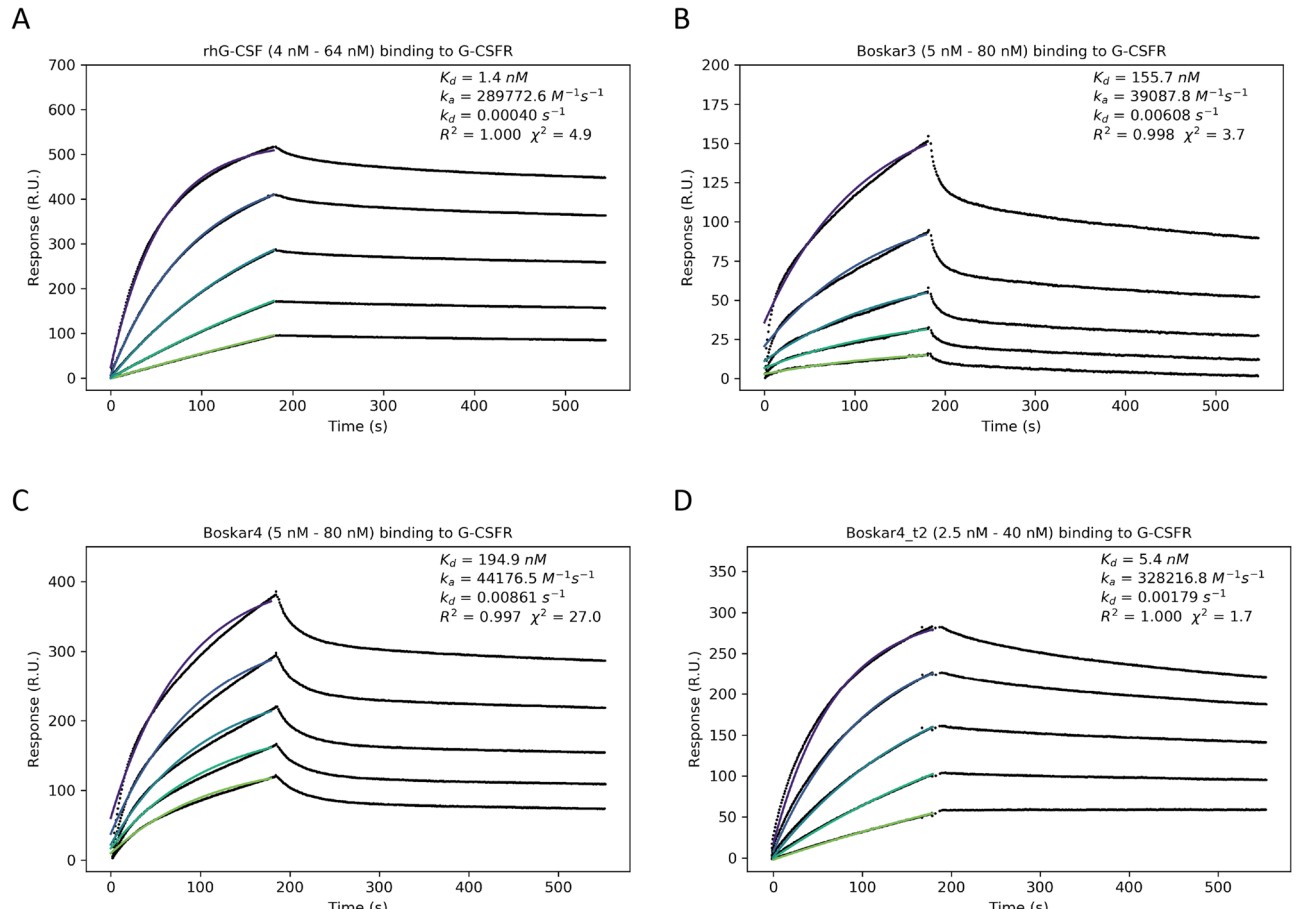

**Fig. 5 The designs bind to the human G-CSF receptor.** SPR titration sensorgrams of the binding kinetics of **A** rhG-CSF (Filgrastim), **B** Boskar3, **C** Boskar4, and **D** Boskar4_t2. The binding kinetics analysis (association curve fits shown in spectrum colours) indicates higher apparent binding affinity of inherently bivalent ligands, whereby rhG-CSF and Boskar4_t2 represent two distinct modes of bivalent binding to the G-CSFR.

**Table 1 SPR binding parameters.**

| Analyte | $k_a$ (M$^{-1}$s$^{-1}$) | $k_d$ (s$^{-1}$) | $K_d$ (M) |
|---|---|---|---|
| rhG-CSF | $(3.0 \pm 0.3) \times 10^5$ | $(4.9 \pm 2.8) \times 10^{-4}$ | $(1.1 \pm 1.6) \times 10^{-9}$ |
| Boskar3 | $(3.8 \pm 0.4) \times 10^4$ | $(6.1 \pm 0.1) \times 10^{-3}$ | $(1.6 \pm 0.2) \times 10^{-7}$ |
| Boskar4 | $(4.3 \pm 0.3) \times 10^4$ | $(8.6 \pm 0.1) \times 10^{-3}$ | $(2.0 \pm 0.2) \times 10^{-7}$ |
| Boskar4_t2 | $(3.4 \pm 0.5) \times 10^5$ | $(1.7 \pm 0.5) \times 10^{-3}$ | $(5.1 \pm 1.9) \times 10^{-9}$ |
| Boskar4_st2 | $(5.3 \pm 0.7) \times 10^5$ | $(2.9 \pm 1.9) \times 10^{-3}$ | $(5.9 \pm 4.4) \times 10^{-9}$ |
| Boskar3_B4 | $(4.8 \pm 0.7) \times 10^5$ | $(1.1 \pm 0.32) \times 10^{-3}$ | $(2.4 \pm 0.9) \times 10^{-9}$ |
| Boskar3_st2 | $(4.9 \pm 0.8) \times 10^5$ | $(2.3 \pm 0.39) \times 10^{-3}$ | $(4.8 \pm 1.4) \times 10^{-9}$ |
| Boskar3 monomeric | $(1.6 \pm 0.1) \times 10^5$ | $(6.3 \pm 0.16) \times 10^{-3}$ | $(3.9 \pm 2.2) \times 10^{-8}$ |
| Boskar3 dimeric | $(1.5 \pm 0.2) \times 10^5$ | $(2.7 \pm 0.19) \times 10^{-3}$ | $(1.8 \pm 3.0) \times 10^{-8}$ |
| Boskar4 monomeric | $(1.4 \pm 0.1) \times 10^4$ | $(1.8 \pm 0.32) \times 10^{-2}$ | $(1.2 \pm 123.3) \times 10^{-6}$ |
| Boskar4 dimeric | $(2.4 \pm 0.2) \times 10^5$ | $(3.7 \pm 0.23) \times 10^{-3}$ | $(1.5 \pm 1.9) \times 10^{-8}$ |

observed where the short-linker tandems are more active than the dimeric species, that is in turn more active than the monomeric species of the respective design (Supplementary Figs. 20 and 21).

**The designs induce proliferation and neutrophilic differentiation of HSPCs in vitro.** To investigate the clinical potential of our designs, we further evaluated their biological activity in human primary CD34$^+$ hematopoietic stem and progenitor cells (HSPCs). We started by assaying the stimulation of CD34$^+$ HSPCs with the original designs (Boskar3 or Boskar4), which induced an increase in cell proliferation that subsequently plateaued, similar to that observed in rhG-CSF-treated cells (Supplementary Figs. 22, 23, and 24). We then assessed granulocytic

differentiation of CD34$^+$ HSPCs from two healthy donors in the presence of rhG-CSF or the different designs using a liquid culture system. In all of the HSPCs experiments, rhG-CSF was used at a concentration of 10 ng/mL, Boskar3 and Boskar4 were used at of 1 μg/mL, and Boskar4_t2 or Boskar4_st2 at 100 ng/mL (except for the CFU and phagocytosis assays where both 10 and 100 ng/ml of Boskar4_t2 or Boskar4_st2 were used). As expected, a majority of cells cultured in the presence of rhG-CSF or designed proteins exhibited the typical and highly specific morphology of neutrophilic granulocytes with multilobed nuclei (Fig. 7A, B). A FACS analysis also revealed differentiation of HSPCs into neutrophilic granulocytes (CD45$^+$CD11b$^+$CD15$^+$, CD45$^+$CD11b$^+$CD16$^+$, and CD45$^+$CD15$^+$CD16$^+$ cells) in the presence of the Boskar3 and Boskar4 to levels comparable to that induced by rhG-CSF (Supplementary Figs. 25 and 26). A more detailed FACS analysis of granulocytic differentiation for Boskar4_t2 and Boskar4_st2 revealed a strong neutrophilic differentiation bias of the treated cells (Fig. 7C, and Supplementary Fig. 27). Additionally, we observed the formation of colony-forming units (CFUs) from healthy donor CD34$^+$ cells in the presence of the designed proteins (Fig. 7D, E, and Supplementary Fig 28). Although the number of CFU colonies induced by the designs was relatively lower than that stimulated by rhG-CSF (Fig. 7D), the typical myeloid cell morphology of CFUs was evident in all groups (Fig. 7E).

We further tested whether the neutrophils differentiated in vitro from HSPCs by our designs can execute neutrophil-specific functions such as production of reactive oxygen species

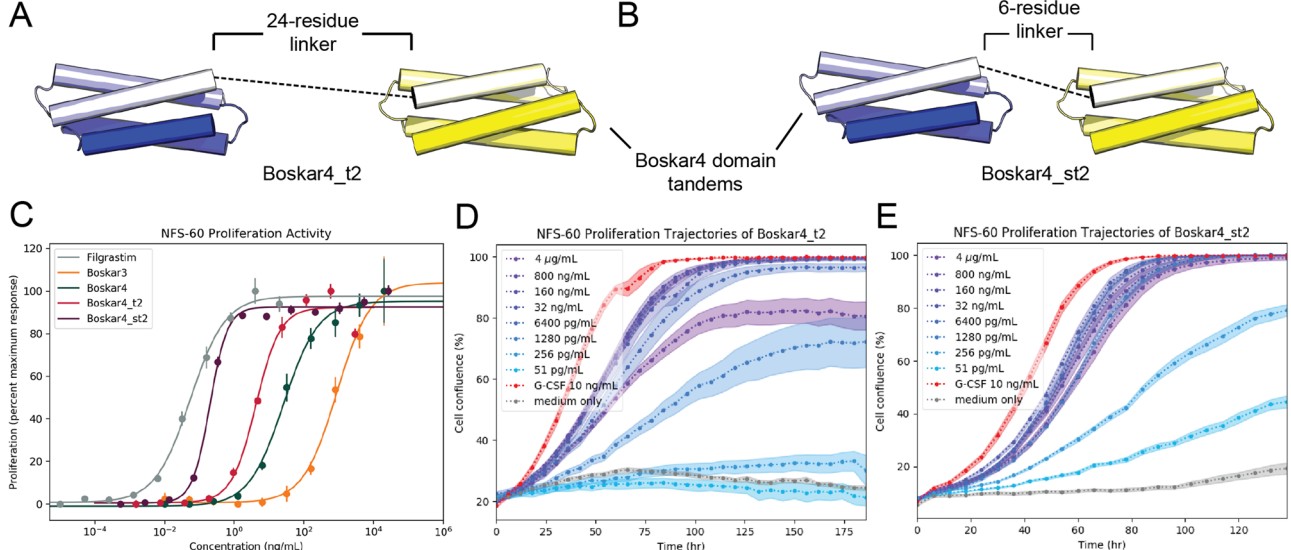

**Fig. 6 The potency of designs can be greatly enhanced by the optimal tandem spacing.** The creation of two-copy Boskar4 tandems with different flexible spacers, through **A** a 24-residue-linker (Boskar4_t2), and **B** a short 6-residue-linker (Boskar4_st2). **C** Dose–response curves in NFS-60 cells show Boskar4_t2 ($EC_{50} = 180$ pM), and Boskar4_st2 ($EC_{50} = 7.6$ pM) to be 11- and 263-fold more active than the single domain Boskar4, respectively. Datapoints and error bars represent mean ± standard deviations of 3 biologically independent replicates. **D, E** This greatly enhanced potency is also evident in the dose- and time-dependent NFS-60 proliferation kinetics, highlighting the requirement for an optimal G-CSFR dimerisation spacing to achieve maximal G-CSFR activation. The experiments in **C, E** were performed as described in figure legend to Fig. 2.

(ROS), neutrophil extracellular traps (NETs) formation, and phagocytosis. We first assessed $H_2O_2$ levels in *N*-Formylmethionyl-leucyl-phenylalanine (fMLP)-stimulated neutrophils and detected elevated ROS levels in designs-generated neutrophils, comparable or even higher to rhG-CSF-stimulated samples (Fig. 7F, and Supplementary Fig. 29). Phagocytosis was evaluated using live cell imaging of neutrophils incubated with pHrodo Green *E. coli* bioparticles. We observed similar phagocytosis behaviour of rhG-CSF- and designs-generated neutrophils (Fig. 7G and Supplementary Fig. 30). These neutrophils were also capable of producing NETs upon stimulation with phorbol myristate acetate (PMA) (Fig. 7H). Taken together, designed proteins induce formation of functionally active neutrophils.

**The design proteins induce chemotaxis of neutrophils in vitro.** To study the effects of design proteins on the chemotactic activity on neutrophils, we assessed chemotaxis of healthy donor peripheral blood neutrophils in vitro by incubating cells with rhG-CSF or the designs and measuring chemotactic activity using IncuCyte® Chemotaxis Cell Migration Assay on an IncuCyte S3 Live cell analysis system. Indeed, we found that our designs induced neutrophils chemotaxis to a similar degree as in with rhG-CSF (Fig. 7I). These results highlight the activity of design proteins not only on the granulocytic differentiation, but also migration of neutrophils.

**Boskar4 tandems induce neutrophil production in zebrafish.** To investigate the potential in vivo activity of our most active design Boskar4_t2 and Boskar4_st2, we used a fluorescent neutrophil reporter zebrafish line, *Tg(mpx:GFP)*[41]. Equal volumes (4 nL) of PBS solution containing 8 ng of rhG-CSF, 16 ng Boskar4_t2, or 16 ng Boskar4_st2 were injected into the cardinal vein of transgenic embryos at 1.5 days post-fertilisation (dpf). As negative control, 16 ng Moevan_control (a protein lacking binding or activation capacity of G-CSFR[28], but expressed and purified under the same conditions) was injected. All injected embryos survived without displaying morphological abnormalities. The number of GFP$^+$ neutrophils was assessed in each

embryo 24- and 72-h post-injection. Compared to the PBS-injected counterparts, the number of neutrophils was almost unchanged in the Moevan_control-injected embryos, whereas it was significantly higher in embryos injected with rhG-CSF, Boskar_t2 and Boskar_st2 treatments (Fig. 8A, B, and Supplementary Table 4). We also observed an increased neutrophil localisation near the caudal hematopoietic tissue in the rhG-CSF and Boskar_st2 injected embryos (Fig. 8C) indicating strong granulopoietic activity for these designs in zebrafish.

**The designed proteins induce granulopoiesis in mice.** We next evaluated the effects of the designed proteins on the induction of granulopoiesis in mice. We treated C57BL/6 mice with rhG-CSF or G-CSF designs, Boskar4_t2 and Boskar4_st2 at a concentration of 300 µg/kg by intraperitoneal injection (i.p.) every second day for a total of five injections. Mice in the control group were treated with PBS, or a protein incapable of G-CSFR binding (Moevan_control[28]) using the same treatment scheme. One day after the fifth injection, the number of Ly6G$^+$Ly6C$^+$ neutrophils in the bone marrow of treated mice was evaluated. We found that treatment of mice with rhG-CSF, Boskar4_t2, or Boskar4_st2 induces production of neutrophils, as compared to the control PBS-treated group (Fig. 8D, and Supplementary Fig. 31). No toxic effects of the designed proteins and no significant effects of Moevan_control on neutrophil numbers were observed. These results demonstrate the granulopoietic activity of our designed proteins in vivo.

**Discussion**
Proteins can uniquely offer large, diverse, and highly restrained physicochemical environments that undergo minimal or tightly controlled modes of motion, as encoded by their primary structure. Cytokines are a class of proteins that exert their function by modulating their cognate receptors in a lock-and-key manner. The latter appears to mostly take place through cytokines dimerising their receptors' ectodomains[42]. Existing structures of helical cytokines undergo minimal conformational change between the receptor-bound and the isolated cytokine structures,

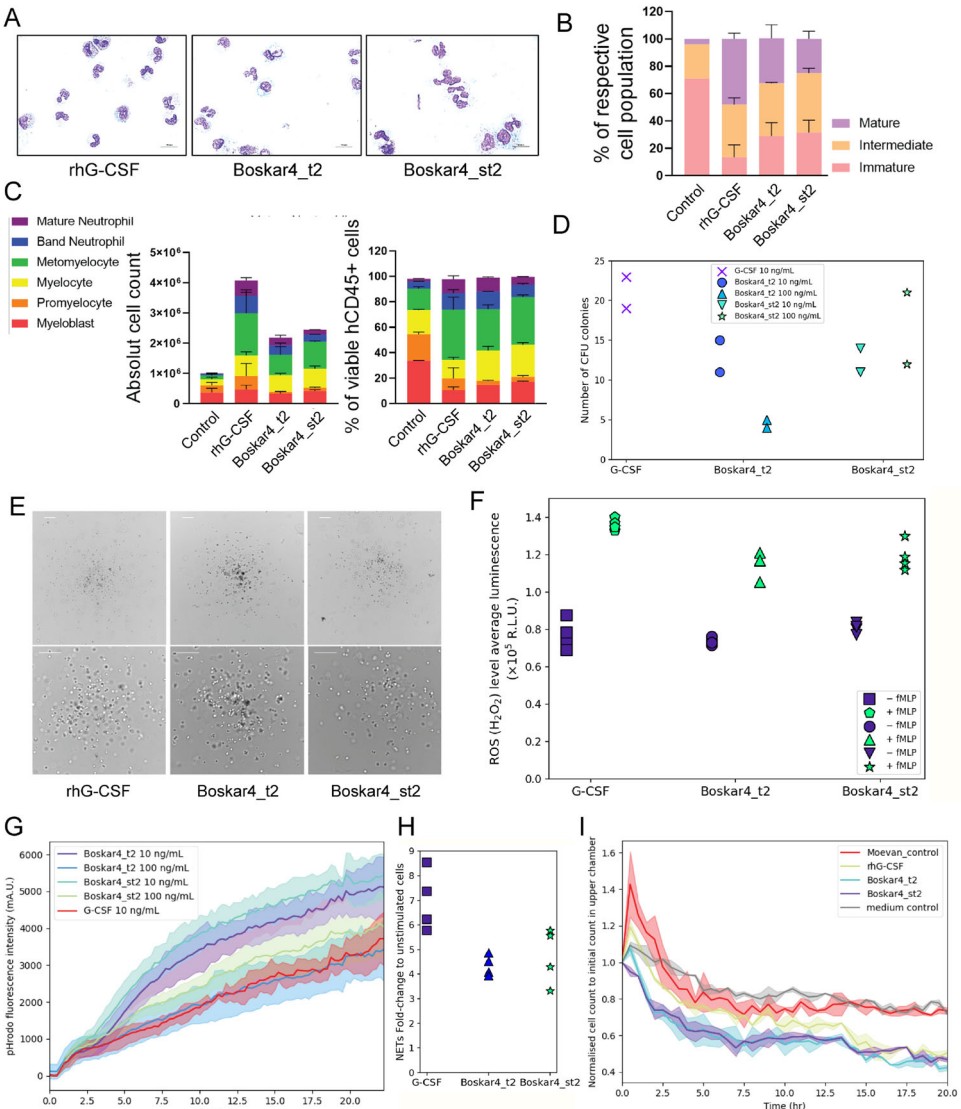

**Fig. 7 The designs are functionally active in human primary cells.** Cell morphology (**A**, **B**) and FACS (**C**) analysis of neutrophils generated from human healthy donors' CD34+ HSPCs in the presence of either the inactive protein Moevan_control (control, 100 ng/ml), rhG-CSF (10 ng/ml), Boskar4_t2 (B4_t2, 100 ng/ml), or Boskar4_st2 (B4_st2, 100 ng/ml) using in vitro liquid culture. Data represent mean ± standard deviation of triplicate samples from two healthy donors. **D**, **E** Quantification (**D**) and representative images (**E**) of myeloid colonies induced by the rhG-CSF (10 ng/ml), Boskar4_t2 (10 or 100 ng/ml), or Boskar4_st2 (10 or 100 ng/ml) from human healthy donors' CD34+ HSPCs after 14 days of culture. **F**–**H** Evaluation of functions of in vitro generated granulocytes using rhG-CSF, Boskar4_t2 or Boskar4_st2 (see the "Methods" section). **F** Reactive oxygen species (ROS) formation in fMLP-stimulated neutrophils. **G** Phagocytosis kinetic analysis using IncuCyte ZOOM System. Lines represent mean, shades represent ± standard deviation. **H** Formation of neutrophil extracellular traps (NETs) was determined by DNA staining in IncuCyte live-cell imaging. The green area was normalised to the phase area at the experiment start, and the results are depicted as fold change from unstimulated to PMA stimulated cells. Data show four biologically independent replicates after 19 h of treatment. **I** Chemotactic migration of peripheral blood healthy donors' neutrophils (solid lines represent the average of duplicates, and shades represent standard deviation) (see the "Methods" section).

indicating a rigid binding mode. However, owing to their regulatory roles, most cytokines are quickly eliminated, and possess poor stability and fairly short serum half-life[43]. The aim of this work was to reconstruct a cytokine to improve its stability and solubility. This was done under the constraint of minimally perturbing the receptor binding epitope, with the goal of preserving the receptor binding affinity, and consequently the underlying pharmacodynamics.

Here, we followed a three-pronged approach of optimising topology, size, and sequence; demonstrating that a functional protein can be fully refactored from around its active epitope. The most critical step in this strategy is the topological rewiring. For instance, a protein architecture with a defined number $N$ of

structural segments (e.g. helices, strands, or motifs), can be rewired into $N(N-1)!$ folds and circular permutations. Thus, the successful design of multiple structured loops is crucial to tap into a huge pool of accessible folds, with distinct biophysical features. We have previously shown a cytokine binding epitope could be rescaffolded onto geometrically accomodating structures with simpler folds[28]. Another recent attempt to redesign a cytokine has utilised a combination of helical extensions and template-based loop design, which successfully yielded highly active Interleukin-2 mimics[44]. The latter strategies, however, are contingent upon finding existing structural templates for either epitope or loop grafting. In contrast, here we deploy a more generalisable approach of de novo loop design that obviates the reliance on

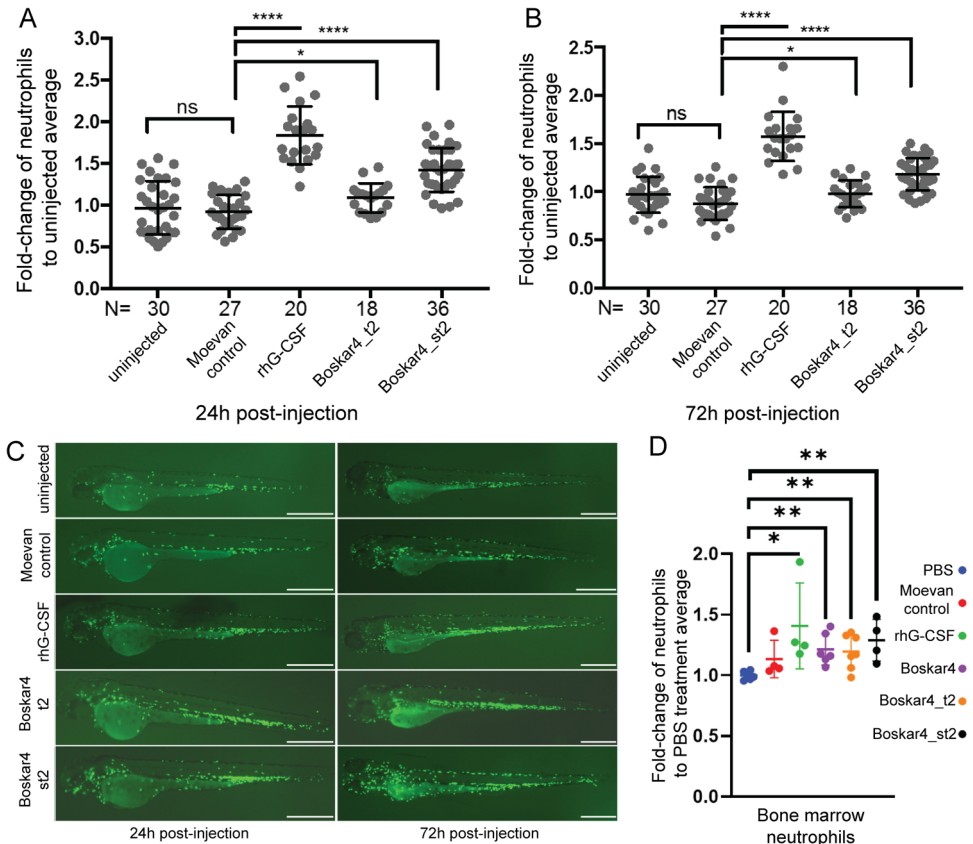

**Fig. 8 The designs show in vivo activity in zebrafish and mouse models. A**, **B** Interval plots of the fluorescent neutrophils in zebrafish (*Tg:mpxGFP*) that were either not injected, or injected with the inactive protein Moevan_control, rhG-CSF, Boskar4_t2, or Boskar4_st2, after (**A**) 24 h or (**B**) 72 h of treatment. Data show mean ± standard deviation, and the number of treated embryos under each condition is show as *N* (under the x-axis), *\*p < 0.05, or \*\*\*\*p < 0.0001 vs. the Moevan_control group (p-values are listed in Supplementary Table 4). **C** Representative images of 1- and 3-day embryos that were not injected, or injected with inactive protein (Moevan_control), Boskar4_t2, or Boskar4_st2. Fluorescently labelled neutrophils can be observed and quantified across the tail region (scale bar: µm). **D** Ly5.1 mice were treated with PBS, the inactive protein Moevan_control, rhG-CSF, Boskar4, Boskar4_t2, or Boskar4_st2 (*n* = 7 per group for each condition). Fold change of mouse neutrophils (DAPI⁻CD45⁺Ly6G⁺Ly6C⁺) in the bone marrow of treated mice compared to PBS group is shown. Statistical significance was calculated by Fisher's two-sided t-test (*\*p < 0.05, or \*\*p < 0.01 vs. the PBS group). The exact p-values are provided in the source data files.

existing loop structures database. Topological rewiring also brings the advantage of eliminating structural segments from across the primary structure, providing an effective means for reducing protein size. The combined effect of topology simplification and size reduction increases the probability of folded contact formation and narrows down the folding landscape, improving the folding rate and free energy, in turn[18,45]. In the last step, core and surface residues must be optimised to maximise conformational stability and solubility. Improving solubility of therapeutic proteins is particularly important to penalise the desolvation energy of off-target interactions, and also to avoid aggregation, which is related to immunogenic effects[46].

We applied our topological refactoring strategy (Fig. 1) to design novel haematopoietic agents that activate the G-CSFR pathway, which induces the proliferation and differentiation of haematopoietic stem and progenitor cells into mature granulocytes. The G-CSF:G-CSFR binding is assumed to occur in a 2:2 complex, through two binding sites on the G-CSF surface; site II and site III[26,47]. Ligand-receptor co-crystal structures, alanine scanning, and NMR studies[26,47–50] point to a minor receptor-binding contribution from site III. Therefore, we sought to design binding domains that maintain the high-affinity binding site II of G-CSF, eliminating site III from the reconstructed designs. Initial cell-based experiments showed nanomolar activity of 2 out of 4 testing designs, bearing only one binding site. Further biophysical

characterisation showed these designs to exist in both monomeric and dimeric forms, whereby the dimeric species were more affine and more active in cell-based assays. This tendency to dimerise of a single binding site-bearing domain appeared to be the reason behind the observed activity. Guided by these results, we set out to create two-copy tandems with different linker lengths to evaluate the impact of purely dimerising ligands with varying receptor subunit spacing. This has surprisingly led us to discover extremely active design variants with low picomolar EC₅₀ (Fig. 6). Our findings are in line with previous findings that the CRH domains in the G-CSF:G-CSFR complex are less than 55 Å apart, highlighting the distance-sensitivity of G-CSFR activation[37]. These findings also motivate further exploration of the G-CSFR subunits sensitivity to their dimeric orientation and spacing, and its impact on activation magnitude and the relative balance of downstream effects (i.e. AKT, ERK, and STAT messaging).

To the end of our design objectives, our biophysical and functional investigations demonstrate that the refactoring strategy enabled the successful creation of novel granulopoietic proteins with sequence and structure starkly distinct from native human G-CSF. The designs have a recombinant expression yield more than 20-fold higher than rhG-CSF, lack any post-translational modifications, and are expressed in a soluble form, unlike rhG-CSF. Moreover, the most active design is highly thermostable, unlike rhG-CSF. The most active design also appears to be more

resistant to proteolytic digestion when compared to rhG-CSF, where Boskar4 digestion is restricted to terminal fragments. Such protease resistance is a feature that may contribute to a potentially longer serum half-life. Deploying this design strategy, a single computational design iteration has shown to be sufficient to yield leads with nanomolar proliferative activity in NFS-60 cells. Additionally, tandem design variants could reach single-digit picomolar activity in the same assay. Combined with the relative proteolytic resistance, such stabilised active proteins can be of great therapeutic potential, especially given that rhG-CSF possesses a very short serum and shelf half-lives. In spite of the stark sequence and structure differences between the designs and native G-CSF, our results from the SPR binding experiments, testing G-CSFR knock-out cells activation, and the analysis of the downstream protein phosphorylation patterns show the designs specifically bind and activate G-CSFR. Detailed functional characterisation of the designs showed very specific activity when used to treat human healthy donor-derived hematopoietic stem cell ex vivo, where the designs induce their differentiation into mature functionally active granulocytes. These design-generated granulocytes were capable of executing a range of functions, such as NETs formation, ROS generation, and phagocytosis. Strikingly, the designs also appear to be pharmacologically active in vivo—in both zebrafish and mouse models. This further highlights the translational power of the computational protein design in creating novel therapeutic leads, without recourse to excessive experimental screening and selection.

Future development of these novel granulopoietic proteins should also evaluate their utility against clinical shortcomings of rhG-CSF. For instance, unmodified rhG-CSF is not orally bioavailable owing to its proteolytic instability[51]. Thus, orally bioavailable formulations of the Boskar designs can greatly improve the quality of life of congenital neutropenia patients that require daily subcutaneous administration of rhG-CSF. Another is the treatment of congenital neutropenia patients with compromising mutations to the Ig-like domain and its binding site in G-CSFR, where such patients are unresponsive to rhG-CSF, which requires two different binding sites to dimerise the receptor, unlike the designs. Conversely, optimising the Boskar designs into strictly monomeric forms can yield G-CSFR signalling inhibitors, that can be useful for different types of cancers[52].

## Methods

**Protein design**. The protein design calculations were done in two stages. The first design stage was the sequence optimisation of the reduced bundle (Fig. 1B), while the second was the design of loops de novo, to rewire the topology (Fig. 1C). The reduced bundle was determined to be the minimal 4-helix-bundle sufficient to scaffold G-CSFR site II; eliminating helix E and three long loops distal to site II. The remaining fragments comprised of the peptide segments 71–123, 10–39, and 145–171, which are made of helices A, B–C and D, where B and C were left connected by a short native loop. This minimal bundle should fully host the essential site II residues, namely: K16, E19, Q20, R22, K23, D27, D109, and D112 (blue patch in Fig. 1A). This helical architecture was reordered to a permutation that would move the native B-C motive to the N-terminus, and wire it with the remaining helices to achieve the shortest gaps (and thus, loops) across the bundle. This transformed the old arrangement: A—E—B–G—D into the new arrangement: B'- C'- A'- D', where "-" and "—" represent short and long loops, respectively. This gapped template was used directly for sequence optimisation (i.e. the first design stage), which was done through sequence and conformer sampling performed within the RosettaScripts framework[53]. With the addition of a root-mean-square deviation (RMSD) constraint on the binding epitope, a previously described core packing protocol was used[54]. This comprised steps of interleaved Monte Carlo sequence, as well as side-chain and backbone conformer sampling iterations. The sequence sampling was directed to most core residues and to solvent-exposed hydrophobic residues. The scoring functions used were the *talaris2013* energy function[55] and the *packstat* packing score[56] (Supplementary methods). While the energy function was used to bias the sampling towards lower energy decoys, the top decoys were forwarded for further evaluation based on the packing quality, where the latter was further judged by the ruggedness of the radial distribution function $g(r)$ as given by the definite integral $\int_0^4 \left| \frac{dg(r)}{dr} \right| dr$.

The second design stage was to construct loops de novo across the junctions C'-A' and A'- D'. The N-C distance gap across the C'- A' junction was approximately 10 Å, while the N-C distance across the A'- D' was approximately 13 Å. Therefore, we sought to build 3- and 4-residue loops at junctions C'- A' and A'- D', respectively. The 3-residue were constructed by covering all sequence combinations of the residues G, D, P, S, L, N, T, E, and K, while the 4-residue loops were constructed by covering all combinations of the residues G, D, P, S, L, N, T, and K. The G, D, P, S, and T amino acids were chosen for their higher propensity to exist in loops[57], L was included due its abundance in helix-helix-turns[58]. E, D and K were included for their helix-capping roles[58], where E was removed from the 4-residue optimisation to reduce the sequence space. Novel loops were constructed through automatic modelling[59], followed by molecular dynamics-based evaluation, which relied on repeated generalised-ensemble sampling (GS)[60] simulations. These simulations were deployed to temper the decoys serially between 250 and 390 K, over 6 and 8 ps, respectively per cycle (through $N = 25$ serial cycles per decoy). These cycles were interlaced with 100 conjugate gradient minimisation steps to revert to local minima along the serial conformational divergence. These simulations were done in restrained and unrestrained formats, initial simulations constrained all atoms except for loop residues, and 5 flanking residues up- and down-stream, later simulations combining top 50 sequences at each loop where recombined and underwent unrestrained simulations. A conformational homogeneity metric was calculated as the average all-against-all RMSD, which is a proxy of the misfolding or basin hopping tendency of each decoy, as $S_h = \sum_{i \neq j}^{N^2} \frac{S_{RMSD_{i,j}}}{N^2}$,

where $S_{RMSD_{i,j}}$ is the aligned backbone RMSD between frames $i, j$. All molecular dynamics simulations relied on conjugate gradient minimisation and a perturbation time step of 2 fs, under Langevin temperature control. A long-range interactions cutoff of 12.0 Å, a switching distance of 10.0 Å, and a Langevin temperature control set to 310 K with a damping coefficient of 1.0 ps$^{-1}$ were used. Generalised Born solvation was used for GS simulations. TIP3P water and Langevin barostat were used for NMR refinement simulations, with a Langevin piston oscillation period of 200 fs and a piston decay period of 50 fs. The systems were neutralised with 0.15 M sodium chloride. All simulations were either conducted using the OpenMM library[61] or the NAMD engine[62]. A schematic of the design workflow is shown in Supplementary Fig. 32.

For computational evaluation of the solubility analysis, we used three different metrics. The first metric was the Damietta solvation energy function, which is based on a CHARMM-compatible solvation model[63]. Using a single round of the Damietta software ra (repack all, v0.22) protocol, residues side chains were repacked, and the residue-wise solvation energy extracted from the output PDB file. The Damietta protein design software is available at https://bio.mpg.de/damietta/. The second metric was the Kyte-Doolittle hydropathy scale[64] as implemented on the ProtScale server (https://web.expasy.org/protscale/). The third metric was the Rosetta residue-wise solvation energy sub-score[65] after a single run of Rosetta Relax protocol using *talaris14* scoring function.

**Protein expression and purification**. Synthetic genes encoding the designs were cloned into the pET28a(+) expression vector carrying a kanamycin resistance gene using NdeI and XhoI cloning sites in-frame with an N-terminal hexaHis-tag and a thrombin cleavage site (Synbio Technologies, Inc.). The plasmids were used to transform chemically competent E. coli BL21(DE3) by heat-shock. Transformed cells were grown in LB medium and induced with IPTG at OD600 of about 0.5–1 with overnight expression at 25 °C. For expression of isotopically labelled protein, a preculture in LB medium was grown, cells collected, washed twice in PBS buffer, and resuspended in M9 minimal medium (240 mM Na$_2$HPO$_4$, 110 mM KH$_2$PO$_4$, 43 mM NaCl), supplemented with 10 μM FeSO$_4$, 0.4 μM H$_3$BO$_3$, 10 nM CuSO$_4$, 10 nM ZnSO$_4$, 80 nM MnCl$_2$, 30 nM CoCl$_2$ and 38 μM kanamycin sulfate, to an OD600 of about 0.5–1. After 40 min of incubation at 25 °C, 2.0 g $^{15}$N-labelled ammonium chloride (Sigma-Aldrich 299251) and 6.25 g $^{13}$C D-glucose (Cambridge Isotope Laboratories, Inc., CLM-1396) were added in a 2.5 l culture. After another 40 min IPTG was added to a final concentration of 1 mM for overnight expression. Cells were collected by centrifugation at 5000 × g for 15 min, lysed by a Branson Sonifier S-250 (Fisher Scientific) in hypotonic 50 mM Tris–HCl buffer supplemented with one tablet of the cOmplete protease inhibitor cocktail (Sigma-Aldrich 4693159001) and 3 mg of lyophilised DNase I (5200 U/mg; Applichem A3778). The insoluble fraction was pelleted by centrifugation at 30,000 × g for 50 min, and the soluble fraction was passed through a 0.45 μm filter and directly applied to a Ni-NTA HisTrap column (GE Healthcare). For wild-type G-CSF, refolding extraction was performed on the insoluble fraction of the E. coli cell pellet by stirring in 8 M guanidinium chloride solution for 2 h at 4 °C. The mixture was gradually diluted to 1 M guanidinium chloride in 4 steps over 4 h and loaded directly onto an affinity purification column. A 5 ml HisTrapFF immobilised nickel column (GE Healthcare Life Sciences 17-5255-01) was used for this purpose, washed consecutively with 30 ml 150 mM NaCl, 30 mM Tris buffer (pH 8.5) at 0, 30 and 60 mM imidazole. Fractions were collected by gradient elution at >60 mM imidazole. The eluate was concentrated using 3 kDa MWCO centrifugal filters (Merck Millipore UFC901024) and loaded onto an equilibrated Superdex 75 gel filtration column (GE Healthcare Life Sciences 17517401). The gel filtration buffer used was 150 mM sodium phosphate buffer, pH 7.5, for NMR and CD transparency as well as cell culture compatibility. An Äkta Pure system (GE Healthcare Life

Sciences) was used for all chromatography runs. All experiments involving wild-type G-CSF were initially conducted using in-house purified rhG-CSF, and later reproduced using USP filgrastim as a standard reference (Sigma-Aldrich 1270435), except the CD experiments, which were performed only using in-house produced rhG-CSF.

**Endotoxin removal and quantification from bacterially produced designs.** We used a special two-step purification protocol to produce low-endotoxin protein samples for in the in vivo experiments. Firstly, we optimised a detergent-based purification protocol based on two previous protocols[66,67]. The cell pellets from overnight expression 2 l cultures were reconstituted in PBS buffer to 25 ml with 3 mg of lyophilised DNase I (5200 U/mg; Applichem A3778) and protease inhibitor cocktail (Sigma-Aldrich 4693159001) and lysed by sonication, as described above. This total lysate was added to chilled 25 ml of 1% Triton-X114 (Sigma Aldrich X114-100ML) and sonicated further at 4 °C for another 30 min interval. This total lysate was then incubated for 5 min in a 42 °C water bath in order to phase out most of the Triton-X114. This was spun at $3000 \times g$, and the supernatant was decanted for further ultracentrifugation at $30,000 \times g$. The supernatant was then loaded on a 5 ml HisTrap NiNTA column, which was prewashed with 0.1% Triton-X114 in PBS (6 column volumes) at 4 °C. After loading, an on-column wash was done with 0.1% Triton-X114 in PBS (10 column volumes), followed by the affinity purification protocol as described above. Size exclusion was performed as described above, with the addition of a pre-equilibration wash with 1 column volume of 3 M urea and 1 M NaOH. Both washes in affinity and size-exclusion purification were flushed through the fraction collector tubing. The second step of the depyrogenation protocol was done using a specialised endotoxin binding resin; the 0.5 ml Pierce high capacity endotoxin removal spin column (ThermoFisher, 88274) according to the supplier's protocol. The amount of endotoxin in protein preparations was quantified using the Pierce chromogenic endotoxin quant kit (ThermoFisher, A39552S) according to the supplier's protocol. A calibration curve was plotted between standard endotoxin concentrations of 0 and 1.0 EU/ml, and samples were diluted to obtain 410 nm absorbance values within this range.

The endotoxin content of the purified designs using this protocol ranged from <0.4 to 6.64 EU/mg, which was considered an acceptable level for in vivo experiments.

**Thermostability analysis.** Circular dichroism (CD) spectra were recorded using a JASCO J-810 spectrometer. Samples (0.5 ml) with concentrations between 0.5 and 0.6 mg/ml of the respective proteins in PBS buffer (pH 7.1) were loaded into 2 mm path length cuvettes. Spectral scans of mean residual ellipticity were measured at a resolution of 0.1 nm across a range of 240–195 nm. The mean residual ellipticity at a wavelength of 222 nm across a temperature range of 20–100 °C (with an increase of 1 °C/min) was tracked in melting curves. Additional thermostability analysis was performed on rhG-CSF to evaluate the influence of disulfide bonds on folding, nanoscale differential scanning fluorimetry (nanoDSF) was performed using Prometheus NT.48 (Nanotemper) and standard Prometheus capillaries (Nanotemper, PR-C002). The same temperature ramp parameters used for CD were set for melting and cooling, using 1 mg/ml protein samples in PBS, in 5 mM Dithiothreitol (DTT) in PBS, and in 2.5 mM reduced glutathione (GSH) and 0.5 mM oxidised glutathione (GSSG) in PBS.

**Protease sensitivity analysis.** Purified human neutrophil elastase was obtained from Enzo Life Science (#BML-SE284-0100). The elastase was reconstituted in PBS buffer (pH 7.1) to a stock concentration of 20 IU/ml. Digestion reactions were conducted in PBS buffer with final concentrations of 300 μg/ml of the tested protein, and 1 U/ml of neutrophil elastase. The reaction mixture was incubated at 37 °C and digestion samples were withdrawn, immediately mixed with SDS sample buffer (450 mM Tris–HCl, 12% glycerol, and 10% SDS) and flash-frozen in a liquid nitrogen bath to stop the reaction after 5, 15 and 30 min from the reaction start. Frozen samples were then heated at 85 °C for 10 min before loading on Novex™ 16% Tricine Protein Gels (Thermo Fisher Scientific #EC6695BOX). Protein gels were incubated overnight in fixing solution (30% ethanol, 10% acetic acid) and then stained using colloidal Coomassie dye (Serva; 35050).

Nano-HPLC–MS/MS was used to analyse the protein bands resulting from the digest of Boskar 4 with Neutrophile elastase. Boskar4 samples incubated with neutrophil elastase (as described above) were separated on SDS-PAGE, where each lane of the gel (16% Tricine protein gel, ThermoFisher Scientific, #EC6695) contains 6 μg Boskar4 digested with neutrophile elastase for 30 min. Excised protein bands were digested in-gel using Arg-C. LC–MS/MS analysis was done on an Easy-nLC 1200 (Thermo Scientific) coupled to a QExactive HF mass spectrometer (Thermo Scientific) as described elsewhere[68]. Peptide mixtures were injected onto the column in HPLC solvent A (0.1% formic acid) at a flow rate of 500 nL/min and subsequently eluted with a 46 min gradient of 5–33% HPLC solvent B (80% ACN in 0.1% formic acid) at a flow rate of 200 nl/min. The seven most intense precursor ions were sequentially fragmented in each scan cycle using HCD fragmentation.

MS data were processed using the MaxQuant software suite v.1.5.2.8[69]. Database search was performed using the Andromeda search engine[70]. Spectra were searched against a Uniprot E. coli database, a database containing 248

commonly observed contaminants and against the sequence of human Neutrophil elastase (UniProtKB-P08246 (ELNE_HUMAN)) as well as a database comprising the sequence of Boskar4 and it's truncated forms. For the latter, all sequences derived from consecutively shortening the amino acid sequence of Boskar4 (including the purification tag), once starting at the N terminus and once at the C terminus, were unified into one database. In the database search, full specificity was required for Arg-C and up to two missed cleavages were allowed. Carbamidomethylation of cysteine was set as fixed modification, protein N-terminal acetylation, and oxidation of methionine were set as variable modifications. False discovery rate (FDR) was set to 1% on protein as well as peptide level. The posterior error probability (PEP) was calculated for each peptide[71].

The proteolytic stability of Boskar4 and rhG-CSF was tested against a spectrum of different proteases including recombinant human Cathepsin D (bio-techne, #1014-AS), recombinant human ADAM10 Protein (bio-techne, #936-AD), Cathepsin L from Human Liver (Merck, #219402), and Cathepsin G from human leucocytes (Merck, #C4428). In detail, 300 μg/ml of Boskar4 (23 μM) or rhG-CSF (μM) were incubated with Cathepsin D (10 μg/ml) in 0.1 M NaOAc, 0.2 M NaCl, pH 3.5, ADAM10 (5 μg/ml) in 25 mM Tris, 2 μM ZnCl₂, 0.005 % (w/v) Brij-35, pH 9.0, Cathepsin L (50 μg/ml) and Cathepsin G (0.5 U/μl) in phosphate buffered saline, pH 7.4, at 37 °C. Samples were taken after 5, 15, and 30 min and the reaction was stopped immediately by addition of SDS sample buffer (450 mM Tris–HCl, 12% glycerol, and 10% SDS) and flash-freezing in liquid nitrogen. For analysis frozen samples were heated at 85 °C for 10 min and loaded on Novex™ 16 % Tricine Protein Gels (Thermo Fisher Scientific, #EC6695). Protein gels were incubated overnight in fixing solution (30% ethanol, 10% acetic acid) and stained using colloidal Coomassie dye (Serva, #35050).

**NMR structure determination.** All spectra were recorded at 310 K on Bruker AVIII-600 and AVIII-800 spectrometers. All spectra were acquired for samples of Boskar4 in PBS buffer with a concentration between 400 and 800 μM, where the measuring temperature throughout (showing best dispersion) was 313 K. Backbone sequential and aliphatic side chain assignments were completed using standard triple resonance experiments (HNCO, HNCA, HNCAB, CC(CO)NH-TOCSY). A CCH-TOCSY experiment was used to aid aliphatic sidechain assignments. The CNH-NOESY for CoMAND analysis was acquired at 800 MHz with a mixing time of 80 ms, and consisted of 112 complex time points in the ¹³C dimension, processed to 256 points in the final matrix. The experimental strips extracted for each residue have been submitted with the PDB entry for the *boskar4* structure. Two further NOESY spectra were acquired at 800 MHz: a ¹³C-HSQC NOESY and a 2D-NOESY spectrum acquired with suppression of signals from ¹⁵N-bound protons, which is intended to isolate contacts to aromatic protons. The latter aided aromatic assignment by linking aromatic spin systems to the respective $C^{\beta}H_2$ protons.

Structures were determined using the CoMAND method, which exploits the high accuracy that can be obtained in back-calculating NOESY spectra with indirect ¹³C dimensions[33]. The CoMAND method involves spectral decomposition of one-dimensional sub-spectra extracted from a 3D-CNH-NOESY spectrum[34]. These sub-spectra (strips) are chosen from a search area centred on assigned ¹⁵N-HSQC positions and thus contain cross-peaks to a specific amide proton. For residues with overlapping search areas, strips were extracted at the estimated maximum for each component, and a scaling factor calculated expressing the contribution of each component to the strip. These 1D strips were decomposed against a library of spectra back-calculated by systematic sampling over a local dihedral angle space, yielding estimates of backbone and side chain dihedral angles for each residue. Although it is possible to decompose even heavily overlapped strips with this procedure[33], we did not do so here, due to the relatively low number of such cases. Later stages of the protocol involve conformer selection aimed at minimising a quantitative R-factor expressing the match between the experimental strips and back-calculated spectra, or a fold-factor designed to isolate the contribution to the R-factor from long-range NOESY contacts[33].

For initial model building, unrestrained Rosetta ab initio[72] folding simulations were performed and generated 10,222 decoys. The corresponding CNH-NOESY spectra of these decoys were back-calculated to evaluate the structure-averaged fold-factors. The decoy with the lowest fold-factor was used to seed five independent unrestrained molecular dynamics simulations. These refinement simulations were carried out using the CHARMM36 force field[73] in explicit solvent using the polarisable TIP3P water model. Trajectories of a total length of approximately 1 μs were run, with frames collected every 100 ps. An initial refined ensemble was compiled through a global greedy minimisation of the R-factor as previously described[33], which converged on a total of 12 frames.

The 12 selected frames represent a thermodynamically relevant ensemble, thermalized to the experiment's temperature (313 K). It may not, however, optimise R-factors well, as this depends on the ability of the MD trajectory to reproduce the distributions of conformers present in the sample, particularly for sidechain rotamers. To optimise R-factors further, we have developed an extended protocol that proceeds via a restrained intermediate ensemble. Restraints for this ensemble were derived from sub-sets of the trajectory selected by greedy R-factor optimisation for single residues. These typically converge on R-factors close to the noise level, and the distributions of backbone dihedrals, sidechain rotamers, H-bond parameters and other distances extracted from these subsets are thus

directly associated with low $R$-factors. This procedure was applied for all 107 observed strips, while jointly optimising any strips with overlapped intensities. Restraints derived in this manner were supplemented by 56 NOE contacts identified using the initial ensemble as a bootstrapping model. These were primarily well-resolved contacts between aromatic sidechains and methyl groups observed either in a $^{13}$C-HSQC-NOESY or in a 2D NOESY acquired with suppression of signals from $^{15}$N-bound protons. The extracted restraints are summarised in Table S2.

The intermediate ensemble was calculated in Xplor-NIH[74] using short, low-temperature simulated annealing runs, cooling from 300 to 100 K via 2000 steps of molecular dynamics in vacuo (timestep 3 fs) with randomised initial velocities. Each of the 12 structures from the initial ensemble was used to calculate five restrained models. Each of these 60 models was then used to seed short, unrestrained simulated annealing runs using a similar protocol, primarily aimed at exploring surface sidechain conformations. Each run yielded a further five models and the intermediate ensemble was formed by pooling the 360 resulting restrained and unrestrained frames. A final set of 17 structures was selected by applying greedy optimisation over all available strips. Accordingly, the selected ensemble had improved $R$-factors relative to the initial ensemble, but can no longer be considered thermodynamically relevant. In this respect it is closer in nature to conventionally calculated structures based on NMR-derived restraints, however it differs in that individual structures are allowed to deviate from restraints. For this reason, restraint violations are not tabulated in Table S2.

For one residue, E23 at the C-terminus of α1, greedy optimisation failed to produce $R$-factors commensurate with the quality of the strip, while factorisation suggested a backbone polymorphism. This had not been resolved during greedy optimisation, as it had not been sampled in Rosetta model building, nor during the MD simulations (Supplementary Fig. 12). The polymorphism was localised to the backbone ψ angle of L22. To accommodate this, the α1–α2 loop was remodelled by restraining L22 to either the canonical helical angles derived from the greedy optimisation (φ/ψ = −56 ± 10°/−31 ± 8°), or the alternate conformer suggested by factorisation ψ = −135 ± 30°. These restraints were applied during further simulated annealing rounds, where the residues of the α1–α2 loop were free (L22–L29) and all other coordinates fixed. Loop models were selected to optimise $R$-factors for residues across the segment. This procedure also identified multiple backbone conformations for S26, despite the low signal-to-noise of this strip.

The $^{15}$N{$^{1}$H}-heteronuclear NOE experiment[75] was acquired at 800 MHz as a pseudo-3D spectrum with interleaving of data points. The reported values are the ratio of the intensity for each residue in an experiment with NOE mixing (mixing time 3 s) versus a reference without mixing. MEXICO water exchange experiments[76] were acquired as a pseudo-3D spectrum at 800 MHz with 5 mixing times (τ) spaced linearly between 50 and 250 ms. The reported values are time constants for backbone amide water exchange processes, calculated for each residue by first subtracting and then normalising by a reference value representing τ = 0. The resulting exponential was fitted after linearising using the polyfit routine in NumPy.

**NFS-60 cell proliferation analysis.** NFS-60 cells were cultured in IL-3-containing RPMI 1640 medium, supplemented with L-glutamine, 10% KMG-5 and 10% FBS (CLS, cell line services). Before each assay, cells were pelleted and washed three times with cold non-supplemented RPMI 1640 medium. After the last washing step, cells were diluted at a density of $6 \times 10^5$ cells/ml in RPMI 1640 containing glutamine and 10% FBS. In order to analyse cell proliferation, NFS-60 cells were grown in the presence of varying concentrations of G-CSF wild-type and designed variants. For this, five-fold dilution series were prepared from stock solutions of wild-type G-CSF (40 ng/ml) and the designs (40 µg/ml) in RPMI 1640 medium supplemented with glutamine and 10% FBS. 75 µl of each dilution were mixed with the same volume of washed cells in a 96-well plate yielding a final cell density of $3 \times 10^5$ cells/ml and G-CSF concentrations varying from 0.00001 to 20 ng/ml for wild-type and 0.01–20,000 ng/µl for the designs. Each 96-well plate contained triplicates of each dilution and the according blanks, including wells containing cells seeded in RPMI 1640 medium supplemented with L-glutamine, 10% KMG-5 and 10% FBS (cls, cell line services) and wells containing medium only. For end-point analysis, following incubation for 48 h at 37 °C and 5% CO$_2$, 30 µl of the redox dye resazurin (CellTiter-Blue® Cell Viability Assay, Promega) was added to the wells, and incubation was continued for another hour. Cell viability was measured by monitoring the fluorescence of each well at a H4 Synergy Plate Reader (BioTek) using the following settings: excitation = 560 ± 9 nm, Emission = 590 ± 9 nm, read speed = normal, delay = 100 ms, measurements per data point = 10. The data were analysed and curves were plotted applying a four-parameter sigmoid fit using SigmaPlot (Systat Software). For time-kinetics analysis, cells were monitored for 136 h in the IncuCyte S3 Live-Cell Analysis System (Essen Bio) with a ×10 objective. Cell proliferation was analysed using IncuCyte S3 Software.

**Surface plasmon resonance (SPR) binding assays.** Multi-cycle kinetics experiments were performed on a Biacore X100 system (GE Healthcare Life Sciences). G-CSF Receptor (G-CSFR) (R&D systems 381-GR-050/CF) was diluted to 50 µg/ml in 10 mM acetate buffer pH 5.0 and immobilised on the surface of a CM5 sensor chip (GE Healthcare 29149604) using standard amine coupling chemistry. The designs and rhG-CSF (USP RS Filgrastim, Sigma-Aldrich 1270435) were diluted in

running buffer (10 mM HEPES, pH 7.4, 150 mM NaCl, 3.4 mM EDTA, 0.005% v/v Tween-20). Analyses were conducted at 25 °C at a flow rate of 30 µl/min. Five sequential 2-fold increasing concentrations of the sample solution (for Boskar3 and Boskar4 from 5 to 80 nM; for Boskar4_t2 from 2.5 to 40 nM; for Boskar4_st2 from 6.5 to 100 nM; for rhG-CSF from 4 to 64 nM) were injected over the functionalized sensor chip surface for 180 s, followed by a 360 s dissociation with running buffer. At the end of each run, the sensor surface was regenerated with a 60 s injection of 10 mM glycine–HCl pH 2.0. The reference responses and zero-concentration sensorgrams were subtracted from each dataset (double-referencing). Association rate ($k_a$), dissociation rate ($k_d$), and equilibrium dissociation ($K_d$) constants were obtained using the linearization method described in ref. [77] and were derived as follows. Global fitting of the association curves to a 1:1 interaction model was performed using the following equation:

$$\Gamma(t) = \Gamma_{GG} - \Gamma_{GG} \cdot e^{-k_{obs} \cdot t} \tag{1}$$

where $\Gamma(t)$ describes the surface load capacity over time ($t$), $\Gamma_{GG}$ is the equilibrium surface load capacity, and $k_{obs}$ is the observed binding rate constant. The previous equation was rewritten as:

$$\Gamma(t) = c + a \cdot e^{-b \cdot t}$$

where the parameters $a$, $b$, and $c$ were fit to the data to minimise the value of $\chi^2$, which is evaluated by the expression:

$$\chi^2 = \frac{\sum(\Gamma_{fit} - \Gamma_{obs})^2}{n - p} \tag{2}$$

where $\Gamma_{fit}$ is the $\Gamma(t)$ function with minimum sum of squared deviations from the observed sensorgram $\Gamma_{obs}$, $n$ is the number of data points ($n = 900$), and $p$ is the number of parameters fitted by optimiser ($p = 3$). The optimisation was performed using the Nelder–Mead method at a tolerance of $10^{-12}$ and a maximum number of $10^6$ iterations. The optimisation bounds for parameters $a$, $b$, and $c$ were $(-7 \times 10^2, 0)$, $(10^{-4}, 10^{-1})$, $(0, 7 \times 10^2)$, respectively. After fitting, the resulting $k_{obs}$ values were plotted against the corresponding analyte concentrations ($C$) to perform a linear regression according to the following equation:

$$k_{obs} = k_a \cdot C + k_d \tag{3}$$

where $k_a$ represents the slope, and $k_d$ represents the $y$-axis intercept of the linear fit. The dissociation constant $K_d$ was calculated as follows:

$$K_d = \frac{k_d}{k_a} \tag{4}$$

To measure the dispersion of a dataset, five additional linear fits of $k_{obs}(C)$ function were performed as described above, but excluding one analyte concentration at a time. Standard deviations of $k_a$, $k_d$, and $K_d$ values were calculated as follows:

$$s = \sqrt{\frac{\sum(x_i - \bar{x})^2}{N - 1}} \tag{5}$$

where $x_i$ is the value of $k_a$, $k_d$, or $K_d$ derived from the $i$th linear fit, $\bar{x}$ is the mean value of $k_a$, $k_d$, or $K_d$, and $N$ is the total number of performed linear fits ($N = 5$).

**CRISPR/Cas9-sgRNA RNP-mediated *CSF3R* KO in NFS-60 cells.** A specific guide RNA (sgRNA) for knock-out of the *CSF3R* gene (cut site: chr4 [+126.029.810: −126.029.810], NM_007782.3 and NM_001252651.1, exon 4, 112 bp after ATG; NP_031808.2 and NP_001239580.1 p.L38) was designed using CCTop at (http://crispr.cos.uni-heidelberg.de)[78]. Electroporation of NFS-60 cells was carried out using the Amaxa nucleofection system (SF cell line 4D-Nucleofector kit, #V4XC-2012) according to the manufacturer's instructions. Briefly, $1 \times 10^6$ cells were electroporated with assembled sgRNA (8 µg) and HiFi Cas9 nuclease protein (15 µg) (Integrated DNA Technologies). Clonal isolation of single-cell derived NFS-60 cells was performed by limiting dilution followed by an expansion period of 3 weeks. Genomic DNA of each single-cell derived NFS-60 clones was isolated using QuickExtract DNA extraction solution (Lucigen #QE09050). PCR was carried out with mouse *CSF3R*-specific primers (forward: 5′-GGCATTCACACCATGGGGCACA-3′, reverse: 5′-GCCTGCGTGAAGCT CAGCTTGA-3′) and the GoTaq Hot Start Polymerase Kit (Promega, #M5006) using 2 µl of gDNA template for each PCR reaction. In vitro cleavage assay was done by adding 1 µM Cas9 RNP assembled by the same sgRNA used for the knock-out experiment to 3 µl of each PCR product. The PCR reactions were incubated at 37 °C for 60 min and run on a 1% agarose gel. The PCR products that showed no cleavage were purified by ExoSAP (ratio 3:1), which is a master mix of one-part Exonuclease I 20 U/µl (Thermo Fisher Scientific, #EN0581) and two parts of FastAP thermosensitive alkaline phosphatase 1 U/µl (Thermo Fisher Scientific, #EF0651). Sanger sequencing of purified PCR products was performed by Microsynth and analysed using the TIDE (Tracking of Indels by Decomposition) webtool[79].

**Evaluation of time-dependent effects of the designs on the proliferation of CD34$^+$ HSPCs and NFS-60 cells.** Cells were incubated in poly-L-lysine-coated

96-well plates ($2 \times 10^4$ cells/well) in an IncuCyte S3 Live-Cell Analysis System (Essen Bio) with a ×10 objective at 37 °C and 5% $CO_2$, and cell proliferation over time was analysed. The CD34$^+$ HSPC cell culture medium contained Stemline II Hematopoietic Stem Cell Expansion medium (Sigma Aldrich; #50192) supplemented with 10% FBS, 1% penicillin/streptomycin, 1% L-glutamine and 1 µg/ml Boskar3, 1 µg/ml Boskar4, or 100 ng/ml rhG-CSF. Cell proliferation was analysed using IncuCyte S3 Software.

**Colony-forming unit (CFU) assay of human HSPCs.** CD34$^+$ HSPCs at a concentration of $1 \times 10^4$ cells/ml were plated in 35 mm cell culture dishes in 1 ml Methocult H4230 medium (Stemcell Technologies) supplemented with 2% FBS, 10 µg/ml 100× Antibiotic–Antimycotic Solution (Sigma) and 50 ng/ml rhG-CSF, or 1 µg/ml Boskar3 or Boskar4. Cells were cultured at 37 °C and 5% $CO_2$. Colonies were counted on day 10–14.

**In vitro granulocytic differentiation of HSPCs.** Human CD34$^+$ HSPCs were isolated from the bone marrow mononuclear cell fraction of two healthy donors by magnetic bead separation using the Human CD34 Progenitor Cell Isolation Kit (Miltenyi Biotech Germany; #130-046-703). CD34$^+$ cells were cultured at a density of $2 \times 10^5$ cells/ml in Stemline II Hematopoietic Stem Cell Expansion medium (Sigma Aldrich; #50192) supplemented with 10% FBS, 1% penicillin/streptomycin, 1% L-glutamine and 20 ng/ml IL-3, 20 ng/ml IL-6, 20 ng/ml TPO, 50 ng/ml SCF, and 50 ng/ml FLT-3L. For liquid culture granulocytic differentiation, expanded CD34$^+$ cells ($2 \times 10^5$ cells/ml) were incubated for 7 days in RPMI 1640 GlutaMAX supplemented with 10% FBS, 1% penicillin/streptomycin, 5 ng/ml SCF, 5 ng/ml IL-3, 5 ng/ml GM-CSF, and 10 ng/ml rhG-CSF, 1 µg/ml of Boskar3 or Boskar4, or 100 ng/ml of Boskar4_t2 or Boskar4_st2 (10 ng/mL of Boskar4_t2 or Boskar4_st2 were also tested in some experiments as indicated in the Fig. 7). Medium was exchanged every second day. On day 7, medium was replaced by RPMI 1640 GlutaMax supplemented with 10% FBS, 1% penicillin/streptomycin and 10 ng/ml rhG-CSF, or 1 µg/ml of Boskar3, or 1 µg/ml of Boskar4, or 10 or 100 ng/ml Boskar4_t2, or 10 or 100 ng/ml Boskar4_st2, or PBS. Medium was exchanged every second day until day 14. On day 14, cells were analysed by flow cytometry on a FACSCanto II instrument using the following antibodies: mouse anti-human CD45 (Biolegend; #304036), mouse anti-human CD11b (BD; #557754), mouse anti-human CD15 (BD; #555402), mouse anti-human CD16 (BD; #561248), and mouse anti-human CD66b (BD; #555724). For all FACS analyses, vital mononuclear cells were selected, and doublets were excluded based on scatter characteristics.

Cell morphology was evaluated on cytospin preparations on day 14 of culture. $10 \times 10^4$ cells per cytospin slide were centrifuged at $400 \times g$ for 5 min at room temperature using a Thermo Scientific cytospin 4 cytocentrifuge. Wright-Giemsa-stained cytospin slides were prepared using a Hema-Tek slide stainer (Ames) and evaluated using a Nikon Eclipse TS 100 microscope with a ×10 objective.

**Assessment of phagocytosis kinetics.** Neutrophils from day 14 of liquid culture differentiation were cultured in RPMI 1640 medium supplemented with 0.5 % BSA and pHrodo Green *E. coli* Bioparticles Conjugate (Essen Bio; #4616) according to the manufacturer's protocol (Essen Bio) at 37 °C and 5% $CO_2$. Briefly, $1 \times 10^4$ cells were seeded in 90 µl medium, and 10 µg of Bioparticles were added to a final volume of 100 µl. The cells were monitored for 8 h in an IncuCyte S3 Live-Cell Analysis System (Essen Bio) with a ×10 objective. The analysis was conducted in IncuCyte S3 Software.

**In vitro reactive oxygen species (ROS) production assay.** Cells were seeded at a density of $1 \times 10^5$ cells/ml with or without 10 nM *f*MLP (Sigma, #F3506) and incubated for 30 min at 37 °C and 5% $CO_2$. The level of hydrogen peroxide ($H_2O_2$), a reactive oxygen species (ROS), was measured with the ROS-Glo $H_2O_2$ Assay kit (Promega, #G8820) according to the manufacturer's protocol.

**Quantitative analysis of NETosis formation.** Ex vivo generated neutrophils were seeded at a density of $2 \times 10^4$ per well in a poly-L-lysin-coated 96-well plate in RPMI without phenol red supplemented with 0.5% BSA and 250 nM IncuCyte® Cytotox Green Dye (Incucyte #4633). NETosis was induced by adding 0,5 µM Phorbol-12-myristate-13-acetate (PMA). Cells were analysed every 30 min at ×20 magnification on the IncuCyte S3 Live cell analysis system. Data analysis was performed using IncuCyte Basic Software, as recommended by the manufacturer.

**Chemotaxis.** Neutrophils were isolated from peripheral blood of healthy donors using EasySep Direct Human Neutrophil isolation kit (StemCell technologies #19666) and resuspended in assay medium RPMI (Gibco, # 72400-054) supplemented with 0.5% FBS (Sigma Aldrich, # A8412-100) at a density of $8.3 \times 10^5$/ml. The pre-cooled IncuCyte ClearView migration plate, including insert and reservoir plate (Essen BioScience, cat. 4582), were coated with recombinant human fibronectin fragment (Takara Clonetech, #T100B) and incubated at room temperature for 60 min. After fibronectin was removed, an insert was placed in a new reservoir plate filled with 200 µl of PBS, and 60 µl of neutrophil cell suspension (5000 cells/well) with or without rhG-CSF, Boskar4_t2 and Boskar4_st2 at a final concentration of 10 ng/ml were loaded into the insert, followed by incubation at room

temperature for 45 min in a sterile condition. After that, the inserts were placed in a fibronectin-coated reservoir plate containing 200 µl/well of rhG-CSF, Boskar4_t2 and Boskar4_st2 at a concentration of 100 ng/ml or control medium (RPMI with 0.5% FBS). The plate was placed into an IncuCyte S3 Live cell analysis system. After 30 min of incubation at 37 °C, recurrence scanning at ×10 magnification for every 45 min was scheduled for 24 h, according to the manufacturer's manual. The IncuCyte's automated algorithms quantified the chemotaxis of neutrophils, using whole-well images of the insert membrane from the top of the ClearView migration plate. The chemotaxis results were reported as a decrease of cell numbers on the top side area of the membrane.

**Neutrophil reporter zebrafish experiments.** The mpx:gfp Zebrafish reporter line was used as a readout for neutrophil number. The embryos were collected at 1.5-day post fertilization (dpf) and anaesthetised with 0.1% MS 222 (tricaine methansulfonate), then injected with 4 nl of Moevan_control (16 ng), rhG-CSF (8 ng), Boskar_t2(16 ng), or Boskar_st2 (16 ng) proteins into the cardinal vein. After the injection by one and three days, embryos were positioned and orientated laterally within cavities formed in 1% agarose on a 96-well plate and imaged by a Nikon-stereomicroscope (SMZ18). Imaris software was used to count the number of neutrophils (GFP-positive cells) in each embryos in the tail region. Fold-change of the cell numbers were calculated by normalising to the average of uninjected mpx:gfp groups.

**Induction of granulopoiesis by designed proteins in C57BL/6 mice.** B6.SJL-PtprcaPepcb/BoyCrl mice (Ly5.1 mice) were maintained under pathogen-free conditions in the research animal facility of the University of Tübingen, according to German federal and state regulations (Regierungspräsidium Tübingen, M 05-20G). Mice, aged between 6 and 8 weeks, were treated with intraperitoneal injections (i.p.) of rhG-CSF, Boskar4, Boskar4_t2, or Boskar4_st2 at a concentration of 300 µg/kg every second day for a total of five injections. Mice were sacrificed one day after the last injection. Mice in the control group were treated with PBS, or Moevan_control using the same schema. Bone marrow cells were isolated by flushing with a 22 G syringe, and filtered through a 0.45 µm cell strainer prior to counting and staining for flow cytometry analyses. Cells were stained with DAPI (Sigma#D9542), anti-mouse CD45.1 PerCP (Biolegend#110725), anti-mouse Ly6C AF488 (Biolegend#128022), and anti-mouse Ly6G APC (Biolegend#127614). Flow cytometry was carried out on a FACS Canto II (BD), and data were analysed using FlowJo (BD). Vital mononuclear cells were selected, and doublets were excluded based on scatter characteristics. Gates were set according to fluorescence minus one (FMO) controls.

**Analysis of signalling effector activation in CD34$^+$ HSPCs.** CD34$^+$ cells were cultured in Stemline II Hematopoietic Stemcell Expansion Medium (Sigma-Aldrich; #50192) supplemented with 10% FBS (Sigma-Aldrich; #F7524), 1% L-glutamine (Biochrom, #K0283), 1% penicillin/streptomycin (Biochrom; #A2213) and a premixed cytokine cocktail containing IL-3 (PeproTech; #200-03), IL-6 (Novus Biologicals; #NBP2-34901), TPO (R&D Systems; #288-TP200), rhSCF (R&D Systems; #255-SC-200) and Flt-3L (BioLegend; #550606). Final concentrations were 20 ng/ml for IL-3, IL-6 and TPO, and 50 ng/ml for SCF and Flt-3L. On day 6 of culture, serum- and cytokine-starved (4 h) CD34$^+$ HSPCs were treated with 20 ng/ml of rhG-CSF, 10 µg/ml of Boskar3 or 10 µg/ml of Boskar4 for 30 or 60 min, fixed in 4% PFA (Merck; #P6148) for 15 min at room temperature, and permeabilised by slowly adding ice-cold methanol (C. Roth; #7342.1) to a final concentration of 90% and incubating for 30 min. Cells were left overnight in methanol at −20 °C and stained on the next day by incubation for 20 min on ice in PBS/2% BSA with specific antibodies recognising the phosphorylated signalling effectors, phospho-Stat3 (Tyr705) (D3A7) XP rabbit mAb (Cell Signalling; #9145); phospho-Stat5 (Tyr694) (C11C5) rabbit mAb (Cell Signalling; #9359); phospho AKT (Thr308) (244F9) rabbit mAb (Cell Signalling; #4056S), and phospho-p44/42 MAPK (Erk1/2) (Thr202/Tyr204) (E10) mouse mAb (Cell Signalling; #9106), or the respective Alexa Fluor 488-conjugated isotype control antibody, anti-mouse IgG (H+L) F(ab')2 fragment (Cell Signalling; #4408) or goat anti-rabbit IgG H+L (Abcam; #ab150077). Thereafter, cells were washed twice in ice-cold PBS/2% BSA and analysed by FACS. The background-corrected fluorescence signal was distinguished from the corresponding phosphorylated proteins by subtracting the fluorescence signal of the appropriate isotype control, estimated at each time point of stimulation, from the specific phospho-protein signal.

**Reporting summary.** Further information on research design is available in the Nature Research Reporting Summary linked to this article.

## Data availability

The data that support this study are available from the corresponding authors upon reasonable request. The NMR structural data generated in this study have been deposited in the Protein Data Bank under the accession code 7NY0, and in the Biological Magnetic Resonance Data Bank under the accession code 34613. Source data are provided with this paper.

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

## Acknowledgements

This research was funded by the Max Planck Society, the DFG, the German Federal Ministry of Education and Research, and the Madeleine Schickedanz Kinderkrebs-stiftung. We thank the European Zebrafish Resource Center for providing *Tg(mpx:GFP) i114* fish. We also acknowledge the Garching Computing Centre of the Max Planck Society for computing time on the COBRA and DRACO supercomputers.

## Author contributions

Conceived and designed the experiments: J.S., B.H.A, M.C., K.M., B.B., K.W., A.L., P.M., M.E. Performed the experiments: B.H.A, M.C., M.R., M.N., J.H., N.A., Y.X., P.Mir, A-.C.K., K.M., K.W.R., M.E. Analysed the data: J.S., B.H.A, M.C., M.R., M.N., J.H., N.A., Y.X., P.Mir, A-.C.K., K.M., B.B., M.E. Contributed materials/analysis tools: J.S., M.C., A.L., P.M., M.E. Wrote the paper: J.S., B.H.A, M.C., N.A., K.W.R., K.M., B.B., K.W., A.L., P.M., M.E.

## Funding

## Competing interests

The designed proteins in this study are included as part of the priority patent application number EP19217185. J.S., B.H.A. and M.E. are the inventors, and the application is filed by MAX-PLANCK-Gesellschaft zur Förderung der Wissenschaften e.V., and Eberhard Karls Universität Tübingen. The other authors declare no competing interests.
