## [Peer Review File · Nature Communications]

A topological refactoring design strategy yields highly stable granulopoietic proteinsReviewers' Comments:

Reviewer #1:

Remarks to the Author:

In this manuscript, Skokowa et al. show an elegant example of protein design by rewiring the granulocyte-colony stimulating factor (G-CSF) to improve expression and stability. The work combines a solid protein design approach with functional validation through a series of assays that show relevant activity in vitro and vivo.

The manuscript is well written with design strategy and validation clearly described.

The methods used are sound and well known, and the statistics and presentation appropriate.

The work is of particular interest both for the protein design community and for researchers who use and study G-SCF, but shows an approach that can be applied potentially to a vast number of targets. Therefore, I recommend accepting the manuscript with minor revisions, upon consideration of few comments and clarifications.

Line 112: Please clarify what does the expression "the most structurally homogenous decoy" means.

Line 121 and materials and methods: "For the 3-residue loop, 9 amino acid types were used to generate 729 (93) combinations, and for the 4-residue loop 8 amino acid types were used 122 to generate 4096 (84) combinations."

Please explain why only 9 or 8 different amino acids types were used and why Glutamate was dropped from the 9amino acid set in one loop.

The computational design protocols should be provided in the supplementary materials.

171 an "L" missing from "right pane" -> right panel

Protease assay: what is the specificity of human neutrophil elastase? Could it be that the lower activity towards boskar4 depends on the lack of cleavage sites rather than only the improved stability?

Monomer-dimer equilibrium: what peak has been used for the functional assays? Monomer or dimer? Do single peaks repartition between monomer and dimer or preserve their state?

The discussion should address some of the discrepancy between different assays. For example, the best design is still 1000 times less potent than the original G-CSF, in terms of EC_{50} for NFS-60 proliferation, and data are qualitatively aligned with K_d estimate with Biacore, with a 100 fold difference. However, when looking at kinases activation, designs and G-CSF are not distinguishable. It would be interesting to comment on these data and what they could possibly indicate in the discussion.

Reviewer #2:

Remarks to the Author:

In their manuscript entitled "A topological refactoring design strategy yields highly stable granulopoietic proteins", Skokowa et al. present two novel granulopoietic proteins designed on the basis of the granulocyte-colony stimulating factor (G-CSF) chosen as a template. By performing several biophysical and functional studies the authors show that the designed proteins exhibit activity in vivo and have the potential to find pharmacological application.

This work overcomes the challenges of de novo protein structure design/prediction by using a topological refactoring strategy to optimize the fold of G-CSF, preserving its active site while simplifying the rest of its scaffold. I think this a beautiful example of protein engineering and I believe

this work can be an important stimulus for the field. However, despite I have no problems to agree with the main conclusions drawn by the authors given the adequate amount of evidence from the functional studies, I have several concerns especially regarding the NMR structure determination part, which I think is more problematic and should be definitely revised and strengthened.

Major concerns:

1. The structure of Boskar4 was determined using the recently developed CoMAND method (Structure 2019), which relies on spectral decomposition and quantification of the agreement between structural models and input spectra. This method, despite may be potentially useful to generate an initial model to be further refined, so far has been proven effective only on Ubiquitin in the original publication and therefore its output cannot be considered as trustable as the authors claim without further validation. Complete reliance on spectral prediction and substitution of thorough NOESY spectra analysis with the use of an R factor can be highly risky in obtaining accurate structural information, especially for heavily overlapping spectral region, which in fact are discarded in the CoMAND analysis. Looking to the submitted PDB report, I was also surprised to read that "The authors did not provide any information on software used for structure solution, optimization or refinement. No chemical shift data was provided. No validations of the models with respect to experimental NMR restraints is performed at this time". You would agree with me that it is really difficult to assess the quality of your structure without presenting any experimental data. Therefore, as a very minimum, I would ask you to do the following:

- Provide a complete list of the acquired NMR experiments and relevant parameters (e.g. resolution, relaxation delay, mixing time in the NOESY experiments). Stating that "Backbone sequential and aliphatic side chain assignments were completed using standard triple resonance experiments" is not enough.
 - Present at least the 2D 1H-15N HSQC spectrum of Boskar4 to show that the protein is well-folded and that the sample is homogenous.
 - Show the key strips of the 3D CNH-NOESY spectrum containing the assigned long-range NOEs (between the different helices of the four-helix bundle) supporting your structural model.
 - Use the assigned backbone chemical shifts to predict the secondary structure of the protein (e.g. by using TALOS+) and compare it with what you would expect based on your protein design.
2. If it would be possible to acquire a 1H-15N HSQC spectrum for the G-CSF, I think the comparison of that spectrum to that of Boskar4 would further strengthen the evidence that site II has been preserved. If so, I would expect the chemical shifts of K16, E19, Q20, R22, K23, D27, D109 and D112 cross-peaks to be very similar in both spectra.
3. With their computational design, the authors aimed to repack the protein core, optimize the solvent-exposed residues and rigidify the binding epitope. I think the degree of success achieved in all these aspects could be easily evaluated experimentally by performing a H-D exchange experiment (followed by NMR) and by measuring 15N backbone dynamics.

Minor concerns:

1. I think the data presented in Fig. 3A-B could be further strengthened by adding a 2D 1H-15N HSQC spectrum of the two proteins acquired before and after the temperature ramp up and ramp down. Rather than relying only on ellipticity, which just provides a global picture, atomic-resolution information would reveal if local minor denaturation occurs.
2. The gel reported in Fig. 3C, especially the region below 10 kDa, should be shown entirely. The hypothesis that partial digestion of Boskar4 is restricted to the (His)6-tag and the TEV cleavage site should be verified, e.g. by repeating the experiment after removal of the purification tags.
3. I think it is absolutely necessary to clarify on the oligomeric states of Boskar3 and 4. By looking to supplementary Figs. 5 and 6 it is obvious that both proteins have the tendency to dimerize, and I assume all your experiments were performed using only the monomeric fraction. However, I think it is important to assess the tendency of the protein to dimerize over time and I would like to see some data in this regard, which may also help explaining the stoichiometry of the SPR binding model.

4. In lines 98-99 the authors said that the human G-CSF contains two disulfide bridges across C69 and C75 and C97 and C107, but I do not see how the position of these cysteines matches to the sequence reported in the supplementary table 1. Please clarify.

Reviewer #3:

Remarks to the Author:

Skokowa et al. use G-CSF as template to design new granulopoietic proteins. The authors went on to show that the proteins they designed, i.e. Boskar3 and 4, were able to induce granulocytic proliferation and CD34+ cells differentiation into functional neutrophils using various in vitro and in vivo models. I have the following comments for authors' consideration:

Major comments:

While the newly generated Boskar proteins appeared to be functional and signaled through G-CSF receptor, they are much less potent as compared to existing recombinant human G-CSF (rhG-CSF). One major point that I find it difficult to understand is that, for the in vitro studies, the authors used rhG-CSF in ng/ml range and the designed proteins in ug/ml range. Curiously, for the in vivo studies the authors used a similar concentration for rhG-CSF and the designed proteins. It is crucial for the authors to discuss what is the rationale behind their approach.

G-CSF is used clinically to treat neutropenia patients, and it acts to stimulate granulopoiesis, as well as mobilizing neutrophils to the circulation. Thus, a key functional assay that is lacking from this study is to examine the mobilizing effect of the Boskar proteins (i.e. mouse model).

Figure 2 A and B, please use the same naming for the recombinant GCSF, please check.

Figure 3C, the authors showed that neutrophil elastase was able to cleave G-CSF effectively but not Boskar4. There is no loading control for the western plot, please include this. In addition, have the authors looked at different types of serine proteases, as well as other classes of proteases, such as cysteine protease, metalloproteases, aspartyl protease?

Figure 5A, G-CSF treated cells in the Methocult assay should form granulocyte-monocyte colonies. However, from the figure, the colonies appeared to be more of CFU-M feature. Can the authors elaborate on this and provide more images to verify this? Can the author also provide explanation as to why the concentration for G-CSF used was in ng/ml range whereas for the Boskar3 and 4 were in ug/ml range?

Figure 5C, It is also important for the authors to perform differential cell count on their cytopsin experiments.

Figure 5D, It is unclear where the CD45+CD16+CD66b+ data were derived from? Based on the Supplemental Figure 11, there was no CD66b staining. Additionally, it is difficult to imagine how the CD45+CD11b+CD15+ and CD45+CD15+CD16+ populations be so similar when quantified as absolute counts when their frequency is quite different. I suggest the authors to re-check the data.

Figure 5I, Gr1 is not a specific marker for neutrophils. Ly6G is a better and a specific marker.

Minor comments:

All supplemental figure legend should be furnished with more information to help reader to understand.

Figure 5D mislabeling of the CD11+b in the figure

Reviewer #4:

Remarks to the Author:

A topological refactoring design strategy yields highly stable granulopoietic proteins

A de novo designed protein, Boskar 4 has been characterized and compared with its template, granulocyte-colony stimulating factor (G-CSF). The authors also report its binding to G-CSF receptor.

Comments:

Subsection titles are not in consistent format. Some are brief (ex: "Biophysical and structural features", while others take sentence-form (ex: "The designs possess granulocytic proliferative potential").

SPR data reveals that protein dimerization appears to be necessary for effective receptor binding. By "minimizing" the design of the protein they may have removed residues that aid in this dimerization. The authors should mention this in the discussion as this is a design criterion that should be taken into account if this strategy were going to be used in the future. In other words, the "excess" residues of the protein do likely still have a structural function that is important for function outside of the active site and this should be mentioned.

Since this work is discussing a new design process that was used, a figure describing the workflow would be helpful. A lot of the terminology used is vague so it is unclear the exact nature of the thought process for refining the proteins.

It's not completely clear how this approach is "physics-based" compared to other computational methods. The authors can clarify by briefly covering the driving forces of other computational methods and how they differ from their approach. This would give more distinction to the physics-based approach being used here.

-line 42: "These properties are functions of a set of controllable parameters: the protein's sequence, topology and size." I can not say this sentence is correct. We don't know what the sequence-function relation in proteins is. Also, there are other parameters that affect protein interactions such as environmental conditions (solution, co-solvents, temperature, PH, ...) and most importantly, we do not know if these are ALL the factors.

-line 47: The introduction stressed on improving 'folding thermodynamics' or 'minimizing the loss of entropy', yet thermodynamic parameters weren't reported. Perhaps perform DSC to report the thermodynamics of the designed variants or reword the paragraphs in the 'Introduction'.

-line 48: please explain what is meant by "topological contact order"?

-line 49-51: please clarify the phrases "simpler domain topologies" and "smaller number of folding intermediates". We are still not able to understand structures of proteins as small as 40 residues (e.g., amyloid-beta in Alzheimer's). The "smaller number" that authors refer to is usually not a distinct number, but an "ensemble" of intermediate structures that a protein undergoes before folding or aggregating and it is extremely difficult (if not impossible) to find all of them.

-line 52-54: in general, excluding the effects of temperature and assuming that heat capacity is constant, the free energy is $dG = dH - T dS$ (G: free energy, H: enthalpy, T: temperature, S: entropy). Saying that decreasing entropy minimizes the folding free energy without specifying all the assumptions and conditions is incorrect. For example, removing long loops has definitely an effect on

the number of hydrogen bonds between protein and water molecules which should be discussed, as well as the nature of amino acids in the removed loops.

-line 76: The authors mention that of rhG-GCSF suffers from "low recombinant production yield, poor solubility and stability, and short shelf- and serum half-lives.." Please report the (values of these) parameters for G-CSF (if there are reported in literature) and the motivation of improving its thermostability. At what concentrations do these cytokines need to be stored? Moreover, the yields are increased at a cost of activity. I believe the novelty of this manuscript lies in de novo design and characterization of Boskar4 (rather than improving the properties of GCSF).

- line 79: Please explain what has been achieved in the mentioned refs by doing classical protein engineering on rhG-CSF. How much have these studies improved the drawbacks of the pharmaceutical features so it is possible to compare it with this manuscript's results at the end.

-lines130-138: Since only %30 sequence identity is conserved please explain what other criteria were used in the designs and which residues were allowed to be designed in each process. Also, please explain what constraints were used on the structure during the designs.

-line130-131: the length of wild type and designed proteins are mentioned to be 174 and 123 amino acids. In Figure 1 they are shown as 163 and 119 in contact maps. Please clarify. Also, I would suggest mapping the binding site on Fig 1A.

The authors mentioned that the refactoring process involved optimization of solvent exposed residues. Is it possible to include hydrophilic plots of G-CSF and Boskar4 or perhaps include the plots of solvent accessible surface area with respect to residue number?

-line 178: "Most of the partial digestion of Boskar4 appears to be restricted to terminal fragments, possibly representing the unstructured hexa-Histidine and TEV cleavage purification tag."-were these observations based on SDS-PAGE gel? Is there Mass Spec to confirm.

-line 327: "minimally perturbing the receptor binding epitope": A quick 15N-HSQC of Boskar4 with GCSF receptor would have helped delineate the residues involved in binding-this would have confirmed the binding epitopes/residues of Boskar 4.

-line 327-8: "goal of preserving the receptor binding thermodynamics"- I'm not sure what experiments support this claim. Perhaps the authors should have performed ITC (instead of SPR) to report the thermodynamic parameters.

-line 340: "here we deploy a more generalisable approach of de novo loop design that obviates the reliance on existing protein structures.": The authors mentioned earlier that they wanted to introduce structural elements to replace loops-which implies that prior structural information was utilized in designing the new variants-pls clarify or there might be something that I'm missing. The grouping of panels into each figure was a little hard to follow. For example, Fig 2E and 2F are discussed after Fig 4. I realize that this may be done due to other factors, but this is something to consider as it does make the article more difficult to follow.

Please include the sequence alignment of G-CSF and all 4 variants of Boskar.

Throughout the manuscript, the authors mentioned that their goal is to 'rigidify' the structure, in particular, the binding epitope. Please provide an explanation as to why relaxation experiments weren't performed; otherwise, pls provide RMSD from MD simulations.

Is there any particular reason of preferring 'median and median absolute deviation' over standard deviation or mean absolute deviation?

The authors should discuss the pM affinity of rhG-GCSF. The rhG-GCSF is 1000-fold active than the designed variants. Include possible reasons why Boskar variants weren't as potent or didn't have high affinity to GCSF receptor.

Pls mention here that the rhG-GCSF is also referred as Filgrastim. This is stated in Figure legend or Methods.

Circular dichroism: is it possible that the lower T_m (of rhG-GCSF) is due to breakage of disulfide bond? It is possible that the absence of disulfide bond may improve thermostability and reversibility of the designed variants. What if a reducing agent is added to the buffer-can it improve the thermostability/irreversibility of the protein?

Pls report the solvent and protein concentration used for NMR.

Table S1 suggests that the wild type is still much more active (lowest EC₅₀). Can the authors comment on this?

Is there evidence that G-CSF of human origin would be active in zebrafish?
I would suggest adding Figure 2F to figure 5.

Minor Revisions (typo's, clarifications, etc.):

- Lines 47-57: While the terminology being used here may be known to people in the field, it may be difficult to follow for the broader scientific community. Consider defining terms such as "topological contact order" or "downhill folding", or describing why these are significant in layman's terms.
- Line 92: Need a reference for this.
- Line 111: Spell out the name for "MD" before using abbreviations.
- Line 130: Edit "were 123 amino acid-long..." to "were 123 amino acids long..."
- Line 136: Spell out the name for "PTM" before using abbreviations.
- Line 137: "...proteins were designed not contain..." to "...proteins were designed not to contain..."
- Table S1: The "50" in "EC₅₀" in the fifth column should be subscripted.
- Supplementary Movies: What were the doses used when treating these cells for the movies?
- Line 163: There is a space after "> " in "> 20 mg/ml". Earlier instance in Line 161 has no space. Either is fine, but be consistent.
- -line 175: "NE enzymatically degrades G-CSF activity": pls remove the word 'activity'.
- Line 254: change "a concentration of 50 nM for designs" to "...for each design"
- Line 288, pls spell out 'dpf'.
- Line 322: "...receptors ectodomains..." to "receptor's ectodomains"
- Line 402: with the computational modeling section, the proteins were modeled in isolation and were significantly less functional than the naturally occurring proteins. Would it be possible to model them binding with the intended receptor to simultaneously optimize both the structure and the binding?
- Line 405: "The reduced bundle was determined to be the minimal helical bundle sufficient..." How was this determined? Is this stated in literature (which would require a citation) or determined by your lab (which should have some discussion on how this was determined).
- Line 459: "cOplete" to "complete"
- Figure S6 legend-typo 'monoric'-pls change it to 'monomeric'

REVIEWER COMMENTS

Reviewer #1 (Remarks to the Author):

In this manuscript, Skokowa et al. show an elegant example of protein design by rewiring the granulocyte-colony stimulating factor (G-CSF) to improve expression and stability. The work combines a solid protein design approach with functional validation through a series of assays that show relevant activity in vitro and vivo.

The manuscript is well written with design strategy and validation clearly described.

The methods used are sound and well known, and the statistics and presentation appropriate.

The work is of particular interest both for the protein design community and for researchers who use and study G-SCF, but shows an approach that can be applied potentially to a vast number of targets.

Therefore, I recommend accepting the manuscript with minor revisions, upon consideration of few comments and clarifications.

Line 112: Please clarify what does the expression “the most structurally homogenous decoy” means.

We have further clarified this expression. We had several occasions in previous projects where rosetta *ab initio* folding resulted in false negative designs, i.e. designs that failed to have funneled folding towards the design model, but were fairly stable in molecular dynamics, and eventually were well-behaved in the lab. Therefore, we found it far more computationally efficient to simulate the “misfolding” of a design, and to test how accessible the nearby energy basins are, using either parallel or serial tempering routines, rather than to try to simply fold it from scratch in rosetta. This *ab initio* folding procedure takes at least a few thousand parallel folding simulations (taking between one to few hours each) before one can tell if a single decoy is showing funneled folding towards its designed coordinates or not.

The conformational homogeneity metric we relied on was an all-against-all RMSD of minimized frames dumped from the end of each tempering cycle (we apply random reinitialisation of velocities between cycles), for a particular decoy. We have provided more detail of this in the respective methods section (lines 508-517).

Line 121 and materials and methods: “For the 3-residue loop, 9 amino acid types were used to generate 729 (93) combinations, and for the 4-residue loop 8 amino acid types were used 122 to generate 4096 (84) combinations.”

Please explain why only 9 or 8 different amino acids types were used and why Glutamate was dropped from the 9 amino acid set in one loop.

This choice was based on the information provided by relatively old papers. As we needed to interrupt the helicity at this region, we chose the residues that are highly probable to be in either a loop- or coil region; this was based on old and new analyses of such abundance by Costantini *et al.* (Biochemical and Biophysical Research Communications, 2006). We also placed an emphasis on including the charged residues that are most likely involved in helix-capping, as well as leucine, which seems to be fairly abundant in helical turns, based on work by Wintjens *et al.* (Journal of Molecular Biology, 1996).

We have justified these choices and included above-mentioned references in the relevant Materials and Methods subsection (lines 502-506).

The computational design protocols should be provided in the supplementary materials.

Done.

171 an "L" missing from "right pane" -> right panel

Both have similar meaning.

Protease assay: what is the specificity of human neutrophil elastase? Could it be that the lower activity towards Boskar4 depends on the lack of cleavage sites rather than only the improved stability?

Zhirong *et al.* (Frontiers in Immunology, 2018; <https://doi.org/10.3389/fimmu.2018.02387>) have shown the hNE substrate specificity to be fairly broad, where the enzyme has no strong sequence bias, apart from a strongly-enriched valine at the cleavage site. Nevertheless, we cannot use this reference to argue with certainty that the fewer valines in Boskar designs are the reason.

In addition to the protease substrate specificity, proteolytic susceptibility also depends on the total and partial unfolding tendency to expose the peptide backbone to the protease active site.

To at least partially answer the reviewer's question, we sought to broaden our proteolytic susceptibility analysis of Boskar4, by comparing G-CSF and Boskar4 against a panel of 4 proteases representing the four major families of proteases; i.e. an aspartyl protease (Cathepsin D), a metalloprotease (ADAM10), a cysteine protease (Cathepsin L), and a second serine protease (Cathepsin G). These results are now presented in the results section and Fig. S7. The results show that both G-CSF and Boskar4 were protease resistant, with the exception of Cathepsin D, which cleaved rhG-CSF, but not Boskar4.

Monomer-dimer equilibrium: what peak has been used for the functional assays? Monomer or dimer? Do single peaks repartition between monomer and dimer or preserve their state?

If it's in dynamic equilibrium, you'll probably get the same distribution again even after SEC. But it's simple and informative either way.

This is a very important point also raised by other reviewers. In answer to this question:

- We pooled both fractions together (i.e. monomeric and dimeric) for all experiments presented for each protein, Boskar3 and Boskar4, since SDS-PAGE analysis showed that they are the same protein.
- As we explain in the results, the high-concentration SPR titrations were suggestive of a 2:2 binding. We have indeed carried out experiments during this revision on isolated fractions of the monomeric and dimeric species. The affinity difference between both species was about 5-fold in case of Boskar4, whereby the dimeric fraction showed higher

affinity. As expected, a dimeric species would have a lower apparent K_d . Analysing the effects of isolated monomer-dimer species of Boskar4 on the proliferation of NFS-60 cells, we found about 10-fold higher activity of the dimer, in comparison to monomer.

- Although Boskar4 (the most active single-domain design) was easy to purify in a single-step Ni-NTA purification, and was majorly monomeric, the monomer-dimer ratios varied significantly between different preparations. Therefore, we sought to construct flexible-linker tandems of Boskar4 as a much cleaner way of constructing and characterising a more efficient dimeriser. We also further investigated the impact of the binding domain spacing. All these new data are included in the revised manuscript.

In response to the point of dynamic equilibrium, we have also attempted to re-run the isolated separate fractions containing either monomeric or dimeric species. upon incubation at room temperature, and it appears that the re-equilibration rates between the dimeric and monomeric species are very low (Fig. S15, and Fig. S16).

The discussion should address some of the discrepancy between different assays. For example, the best design is still 1000 times less potent than the original G-CSF, in terms of EC_{50} for NFS-60 proliferation, and data are qualitatively aligned with K_d estimate with Biacore, with a 100 fold difference. However, when looking at kinases activation, designs and G-CSF are not distinguishable. It would be interesting to comment on these data and what they could possibly indicate in the discussion.

The difference in the granulopoietic activity between G-CSF and Boskar4 might be explained by the mixture of less active monomers and more active dimers of Boskar4 in protein preparation. Our new variants of Boskar4, Boskar4_t2 and Boskar4_st2, are however much more active: The EC_{50} values of Boskar4_t2 and Boskar4_st2 were far lower and accordingly we were able to successfully differentiate healthy donor CD34+ cells into functional neutrophils by either 100 ng/mL or 10 ng/mL of these tandem repeat designs of Boskar4. We added these new data in the revised manuscript.

We moved the old figures of the differentiation and functional assays results in the revision for Boskar3 and Boskar4 (Figures S18-22, S24-26), to make room in the main figures to explain the reasoning behind the design tandems and show their functional properties (Fig. 5-8). We also added another clarification of the concentrations used in the Results section. We indicate in Fig. 7 when the different concentrations of the design were used.

Reviewer #2 (Remarks to the Author):

In their manuscript entitled "A topological refactoring design strategy yields highly stable granulopoietic proteins", Skokowa et al. present two novel granulopoietic proteins designed on the basis of the granulocyte-colony stimulating factor (G-CSF) chosen as a template. By performing several biophysical and functional studies the authors show that the designed proteins exhibit activity in vivo and have the potential to find pharmacological application.

This work overcomes the challenges of de novo protein structure design/prediction by using a topological refactoring strategy to optimize the fold of G-CSF, preserving its active site while simplifying the rest of its scaffold. I think this a beautiful example of protein engineering and I believe this work can be an important stimulus for the field. However, despite I have no problems to agree with the main conclusions drawn by the authors given the adequate amount of evidence from the functional studies, I have several concerns especially regarding the NMR structure determination part, which I think is more problematic and should be definitely revised and strengthened.

Before answering the detailed point-by-point remarks raised by the reviewer, we understand the reviewer's main concerns are centred on the CoMAND method of determining the Boskar4

solution structure. While we acknowledge that the method is not yet widely adopted, we maintain that it is ideally suited to protein design projects. Firstly, it probes local conformational diversity at a very detailed level, providing direct feedback on the success of the design. Secondly, in such projects it is important to have a “white-gloves” method, given that the design model inevitably represents a strong conformation bias. That said, part of the reason for proposing CoMAND was to explore the possibility of obtaining thermodynamically relevant ensembles, thermalized at the measurement temperature of the spectra. It is very difficult to obtain such an ensemble in conventional NMR structure determination via restrained simulated annealing, as the force constants on covalent geometry terms must be kept strong relative to those of restraints. This result is in covalent geometry more in keeping with cryogenic crystal structures than with a protein in solution at room temperature. However, thermodynamic relevance may come at the cost of further optimizing R-factors, and we concede that here low R-factors are a higher priority, as they provide confidence in the integrity of the obtained structure.

To this end we have recalculated Boskar4 solution structure using an extended protocol, designed in part following reviewer comments on the original paper. In this protocol, we use greedy optimization to obtain sub-structures that minimize R-factors for individual residues. These local conformations are then used to extract restraints that are used to calculate an intermediate ensemble. This intermediate ensemble provides starting structures for short, unrestrained MD simulations, generating a pool of structures from which the final ensemble is selected. This protocol has the advantage that this pool is weighted toward conformers known to contribute to low R-factors, providing better conditions for selection. Also, as the intermediate ensemble is calculated using restraints, any conventional NMR data can be added at this stage. We have chosen to add distances restraints from NOE contacts between methyl groups and aromatics in the protein core, that are readily assigned. The new ensemble is both better focused and has better R-factors than the original one, which will no longer be submitted. We have added descriptions of this protocol to the materials and methods, and extended the analysis in the results section. We also provided the details of the restraints used to calculate the intermediate ensemble in Supplementary Table S2, S3.

Major concerns:

1. The structure of Boskar4 was determined using the recently developed CoMAND method (Structure 2019), which relies on spectral decomposition and quantification of the agreement between structural models and input spectra. This method, despite may be potentially useful to generate an initial model to be further refined, so far has been proven effective only on Ubiquitin in the original publication and therefore its output cannot be considered as trustable as the authors claim without further validation.

- The original paper on the CoMAND method was aimed at demonstrating the basic strategy of conformational mapping by NOESY decomposition. To this end, we carried out much of the detailed analysis on human Ubiquitin, as this represents a gold standard in NMR structure determination and many ensembles of very high quality are available. However, we also demonstrated the method to be broadly applicable using four further proteins with a diverse range of topologies and secondary structure content, including a helical bundle analogous to the current design (polb4). We believe that this represents a reasonable validation set. We have since added several further structure determinations, which will be the subjects of upcoming publications. While we agree with the reviewer that ubiquitin may represent “low-hanging fruit” for NMR papers describing new experiments, it is hardly so for structure determination, where the wealth of existing ubiquitin ensembles obtained by different methods sets a very high standard. That CoMAND was able to compile a thermodynamically relevant ensemble that explained the input spectra better than existing ensembles can be considered a strong validation of the method.

Complete reliance on spectral prediction and substitution of thorough NOESY spectra analysis with the use of an R factor can be highly risky in obtaining accurate structural information, ...

- While we are aware that previous attempts to incorporate quantitative R-factors in NMR structure determinations have met with little success, none of these is directly comparable to CoMAND. Firstly, we are unaware of any previous method that has used analytical decomposition to obtain local conformational parameters. Secondly, we have introduced several elements, such as use of indirect ^{13}C -dimensions, which are key to the accuracy of back-calculation. Under these conditions, we are able to use R-factor optimisation to explain most strips to the noise level. We maintain that this should extract the maximum structural information from the data. In contrast, conventional NOESY analysis relies on peak-picking to interpret the spectra and compiles a set of restraints based on this interpretation. The analysis we presented in the CoMAND paper suggests that this process results in considerable loss of information, such that structures generated from those restraints often poorly explain the input spectra. It is thus open to discussion which of the two methods represents a more “thorough” analysis.

... especially for heavily overlapping spectral region, which in fact are discarded in the CoMAND analysis.

- We did not intend to imply that overlapped strips were discarded from the analysis entirely, only that they were discarded from the initial model building step. We often omit overlapped strips in the initial factorization, as the following Rosetta routines can easily accommodate short gaps in residue coverage. We have rewritten this section to make this distinction clear. In fact, CoMAND copes well with cases of overlap, as we demonstrated in Fig. 3C of the CoMAND paper. Moreover, “greedy” optimization treats overlapped strips implicitly and we have now added improved methods for calculating the R-factor for overlaps using scaling terms. Thus CoMAND analysis treats overlaps at least as well as conventional analysis.

Looking to the submitted PDB report, I was also surprised to read that “The authors did not provide any information on software used for structure solution, optimization or refinement.

We acknowledge this issue, and we appreciate that much detail that would usually be associated with a PDB structure deposition was missing. This issue is related to the nature of the PDB deposition system, which expected files in forms and formats not relevant or available for CoMAND structure determination. Without these required items, the deposition was treated as incomplete and even routine processes, such as chemical shift submission, were stalled by the PDB webforms. The PDB deposition system now recognizes CoMAND as a structure determination method and the updated ensemble could therefore be – and has been – submitted normally, with all of the associated data (as PDB: 7NY0, BMRB: 34613).

No chemical shift data was provided. No validations of the models with respect to experimental NMR restraints is performed at this time”. You would agree with me that it is really difficult to assess the quality of your structure without presenting any experimental data.

We have added a line in the Materials and Methods to note that experimental CNH-NOESY strips have been deposited with the PDB entry.

Therefore, as a very minimum, I would ask you to do the following:

- Provide a complete list of the acquired NMR experiments and relevant parameters (e.g. resolution, relaxation delay, mixing time in the NOESY experiments). Stating that “Backbone sequential and aliphatic side chain assignments were completed using standard triple resonance experiments” is not enough.
- Present at least the 2D ^1H - ^{15}N HSQC spectrum of Boskar4 to show that the protein is well-folded and that the sample is homogenous.
- Show the key strips of the 3D CNH-NOESY spectrum containing the assigned long-range NOEs (between the different helices of the four-helix bundle) supporting your structural model.

- Use the assigned backbone chemical shifts to predict the secondary structure of the protein (e.g. by using TALOS+) and compare it with what you would expect based on your protein design.

We have added more detail on the NMR experiments used, an assigned ^{15}N -HSQC figure, and a supplementary table detailing inter-helical contacts, as requested. We had previously stated that the factorization results agree with TALOS-based predictions. We have added a supplementary figure detailing this comparison. We have also added more detail on the structure to the results section, including a description of a structural polymorphism in the capping motif for $\alpha 1$, which should provide an impression of the level of detail available in CoMAND analysis. This information can be found in the revised Results and Materials and Methods sections, Supplementary tables S2 and S3, updated Fig. 4, and Fig. S8-11.

2. If it would be possible to acquire a ^1H - ^{15}N HSQC spectrum for the G-CSF, I think the comparison of that spectrum to that of Boskar4 would further strengthen the evidence that site II has been preserved. If so, I would expect the chemical shifts of K16, E19, Q20, R22, K23, D27, D109 and D112 cross-peaks to be very similar in both spectra.

Although this is good idea, especially as the shifts of a relevant G-CSF construct have been published and further measurement is not even necessary, the sequence of Boskar4 differs considerably from the parent, such that stretches of more than a few amino acids without a point mutation are rare along the Boskar4 sequence. We tried to make a comparison, as reviewer suggested, but it was not conclusive.

3. With their computational design, the authors aimed to repack the protein core, optimize the solvent-exposed residues and rigidify the binding epitope. I think the degree of success achieved in all these aspects could be easily evaluated experimentally by performing a H-D exchange experiment (followed by NMR) and by measuring ^{15}N backbone dynamics.

This is another good idea. For the water exchange, we prefer to directly measure amide proton exchange rates via MEXICO-type experiments, which can provide exchange constants even on millisecond timescales. We have also acquired a $^{15}\text{N}\{^1\text{H}\}$ -heteronuclear NOE experiment, thus covering both the fastest and slowest timescales of internal motion. These new data are now presented in Fig. 4D.

Minor concerns:

1. I think the data presented in Fig. 3A-B could be further strengthened by adding a 2D ^1H - ^{15}N HSQC spectrum of the two proteins acquired before and after the temperature ramp up and ramp down. Rather than relying only on ellipticity, which just provides a global picture, atomic-resolution information would reveal if local minor denaturation occurs.

We already have CD spectra before and after the temperature ramps. We also have 2D HSQC spectra acquired before and after several days of acquisitions at 313 K (40 °C), where we did not observe changes over this period at that temperature. Therefore, we do not think the suggested experiment will add much information.

2. The gel reported in Fig. 3C, especially the region below 10 kDa, should be shown entirely. The hypothesis that partial digestion of Boskar4 is restricted to the (His)₆-tag and the TEV cleavage site should be verified, e.g. by repeating the experiment after removal of the purification tags.

This is an important remark. As the efficiency of the thrombin cleavage was not so high and that Boskar4 tends to dimerise, we sought to perform mass spectrometry of NE-digested Boskar4 to analyse the composition of the observed bands. Our results indicate it is both N- and C- terminal degradation of a small stretch of the sequence on either side (Fig. S6). These satellite bands were absent as we tested the degradation of Boskar4 by a panel of diverse proteases (Fig. S7),

which may suggest that this terminal degradation by hNE is perhaps sequence specific (but we prefer not to make claims in that direction).

3. I think it is absolutely necessary to clarify on the oligomeric states of Boskar3 and 4. By looking to supplementary Figs. 5 and 6 it is obvious that both proteins have the tendency to dimerize, and I assume all your experiments were performed using only the monomeric fraction. However, I think it is important to assess the tendency of the protein to dimerize over time and I would like to see some data in this regard, which may also help explaining the stoichiometry of the SPR binding model.

We agree with the reviewer that this is a critical point. As we have clarified above to reviewer #1, we have aimed to query this oligomeric state-activity relationship in different ways. To summarise again: We looked at the binding model, tested the affinity and activity of the isolated fractions, analysed the interchange rate between the monomeric and dimeric species, and we improved our designs' activity by more than 2 orders of magnitude by creating differently spaced covalent tandems. This new information has been added to Fig. 5- 8, and Fig. S16 and S17 of the revised manuscript.

4. In lines 98-99 the authors said that the human G-CSF contains two disulfide bridges across C69 and C75 and C97 and C107, but I do not see how the position of these cysteines matches to the sequence reported in the supplementary table 1. Please clarify.

We have now clarified that the sequence numbering used in the text is the canonical Swiss-Prot numbering of Human CSF3. We also indicated that the sequence shown in Table S1 is the structured part of PDB 2D9Q, where we only considered the structured part of the complete G-CSF sequence (which is 7 residues shorter than the FDA approved form; <https://go.drugbank.com/drugs/DB00099>). We accordingly highlighted these cysteine residues in Table S1.

Reviewer #3 (Remarks to the Author):

Skokowa et al. use G-CSF as template to design new granulopoietic proteins. The authors went on to show that the proteins they designed, i.e. Boskar3 and 4, were able to induce granulocytic proliferation and CD34+ cells differentiation into functional neutrophils using various in vitro and in vivo models. I have the following comments for authors' consideration:

Major comments:

While the newly generated Boskar proteins appeared to be functional and signaled through G-CSF receptor, they are much less potent as compared to existing recombinant human G-CSF (rhG-CSF). One major point that I find it difficult to understand is that, for the in vitro studies, the authors used rhG-CSF in ng/ml range and the designed proteins in ug/ml range. Curiously, for the in vivo studies the authors used a similar concentration for rhG-CSF and the designed proteins. It is crucial for the authors to discuss what is the rationale behind their approach.

As mentioned in our comments to reviewers #1 and #2, we generated Boskar4_t2 and _st2 variants, which are tandem repeats of Boskar4, having much better activity than Boskar4 rhG-CSF. The difference between the effects of Boskar4 *in vitro* and *in vivo* might be explained by improved stability and protease resistance of Boskar4, as compared to rhG-CSF. Enhanced stability and protease resistance are essential features for the *in vivo* activity of proteins.

G-CSF is used clinically to treat neutropenia patients, and it acts to stimulate granulopoiesis, as well as mobilizing neutrophils to the circulation. Thus, a key functional assay that is lacking from this study is to examine the mobilizing effect of the Boskar proteins (i.e. mouse model).

We tested in vitro chemotactic/mobilizing activity of the designs on peripheral blood neutrophils isolated from healthy donors. We detected comparable or even better chemotactic effects of the

design proteins as compared to rhG-CSF. We added these new data in the Fig. 7I of the revised manuscript.

Figure 2 A and B, please use the same naming for the recombinant GCSF, please check. We changed the naming to rhG-CSF throughout the manuscript.

Figure 3C, the authors showed that neutrophil elastase was able to cleave G-CSF effectively but not Boskar4. There is no loading control for the western plot, please include this. In addition, have the authors looked at different types of serine proteases, as well as other classes of proteases, such as cysteine protease, metalloproteases, aspartyl protease?

The experiment in Fig. 3C was visualised as a Coomassie gel (we did not do it as a western blot). This experiment was with samples from the same stock of purified protein, which is why a loading control is not needed in this experiment.

As per the reviewer's suggestion, we have performed proteolysis assays using a representative protein from each of the four major families of proteases - the lysosomal aspartyl protease Cathepsin D, the metalloprotease ADAM10, the cysteine protease Cathepsin L, and the serine protease Cathepsin G. We found that both Boskar4 and rhG-CSF were not sensitive to Cathepsin G, Cathepsin L or ADAM10. However, Cathepsin D cleaved rhG-CSF, but not Boskar4. We have added these findings to the Results and Fig. S7.

Figure 5A, G-CSF treated cells in the Methocult assay should form granulocyte-monocyte colonies. However, from the figure, the colonies appeared to be more of CFU-M feature. Can the authors elaborate on this and provide more images to verify this? Can the author also provide explanation as to why the concentration for G-CSF used was in ng/ml range whereas for the Boskar3 and 4 were in ug/ml range?

We made new CFU images with better resolution, where CFU-GM colonies are seen. We further improved the design of Boskar4 to make it more active.

As mentioned above, our new designs Boskar4_t2 and Boskar4_st2 variants are much more active than Boskar4. We produced mature functionally active neutrophils *in vitro*, stimulating CD34+ HSPCs with 100 ng/ml of Boskar4_t2, or_st2. In the CFU assay, new designs were active at 10 ng/ml concentration.

Figure 5C, It is also important for the authors to perform differential cell count on their cyospin experiments.

We performed differential cell counts of cyospin preparations. is the results are included in Fig. 7C,D of the revised manuscript.

Figure 5D, It is unclear where the CD45+CD16+CD66b+ data were derived from? Based on the Supplemental Figure 11, there was no CD66b staining.

We corrected supplemental figure 21.

Additionally, it is difficult to imagine how the CD45+CD11b+CD15+ and CD45+CD15+CD16+ populations be so similar when quantified as absolute counts when their frequency is quite different. I suggest the authors to re-check the data.

The data are correct. The total number of cells at day 14 of differentiation was different in each well/stimulation condition. We calculated absolute cell numbers based on the total number of cells. Additionally, suppl. Figure 21 is showing representative FACS blots of one well per stimulation condition, and the percentage of respective cell populations in the second well may slightly vary.

Figure 5I, Gr1 is not a specific marker for neutrophils. Ly6G is a better and a specific marker.

As the reviewer suggested, we used Ly6G and Ly6C in the analysis of new in vivo data, to be able to better discriminate between neutrophils and monocytes/eosinophils.

Minor comments:

All supplemental figure legend should be furnished with more information to help reader to understand.

We added more information to supplemental figure legends.

Figure 5D mislabeling of the CD11+b in the figure.

We corrected mislabeling. It is Fig S22 now.

Reviewer #4 (Remarks to the Author):

A topological refactoring design strategy yields highly stable granulopoietic proteins
A de novo designed protein, Boskar 4 has been characterized and compared with its template, granulocyte-colony stimulating factor (G-CSF). The authors also report its binding to G-CSF receptor.

Comments:

Subsection titles are not in consistent format. Some are brief (ex: "Biophysical and structural features", while others take sentence-form (ex: "The designs possess granulocytic proliferative potential").

We have rewritten the subsection titles in a uniform style now.

SPR data reveals that protein dimerization appears to be necessary for effective receptor binding. By "minimizing" the design of the protein they may have removed residues that aid in this dimerization. The authors should mention this in the discussion as this is a design criterion that should be taken into account if this strategy were going to be used in the future. In other words, the "excess" residues of the protein do likely still have a structural function that is important for function outside of the active site and this should be mentioned.

The reviewer raises an important point. While there is evidence in the literature that a second binding site (or the so called *site III*) exists, the location of that "epitope" on the G-CSF surface and its "paratope" on the G-CSFR Ig-like domain are not very clear. So we assumed that the high-affinity binding site (*site II*) may be the only site needed for activation. This assumption is based on studies suggesting that cytokine receptors pre-exist as constitute dimers and are rather conformationally rearranged upon ligand binding (Tenhumberg et al., Biochemical and Biophysical Research Communications, 2006, Constantinescu, et al., Proceedings of the National Academy of Sciences, 2001, and Waters and Brooks, Biochemical Journal, 2015), in addition to structural studies (PDB: 2D9Q, 1CD9).

The question raised by the reviewer falls in line with the central query of the bivalent binding and the putative mode of receptor activation. We have put considerable effort into querying this through multiple experiments, as explained above and in response to the other reviewers. This eventually led us to construct and test tandem chimeras which were much more potent activators than the initial designs. Moreover, manipulating the flexible spacer length greatly influences the activity outcome (i.e. Boskar4_t2 vs Boskar4_st2), without an impact on the affinity to the receptor.

In conclusion, our eventual findings are strongly supporting the “bivalent binding” mechanism, but they also corroborate the findings of Mine, *et al.* (Biochemistry, 2004) that the ideal active configuration requires an inter-CRH domain distance substantially shorter than 55 Å. This, and the fact that related cytokine receptors (particularly TPOR and EPOR) have been shown to be sensitive to the exact dimeric orientation, shall guide our further designs that aim to achieve smaller and more active receptor agonists. We have rewritten the discussion to address these points.

Since this work is discussing a new design process that was used, a figure describing the workflow would be helpful. A lot of the terminology used is vague so it is unclear the exact nature of the thought process for refining the proteins.

We added Fig. S28, which summarises the design workflow. We have also added examples of the design scripts used (see Supplementary Methods), and added further details in the Materials and Methods section.

It's not completely clear how this approach is “physics-based” compared to other computational methods. The authors can clarify by briefly covering the driving forces of other computational methods and how they differ from their approach. This would give more distinction to the physics-based approach being used here.

We meant that physics-based protein design offers advantages of empirically guided engineering approaches. We have removed the term “physics-based” to avoid the confusion.

-line 42: “These properties are functions of a set of controllable parameters: the protein's sequence, topology and size.” I can not say this sentence is correct. We don't know what the sequence-function relation in proteins is. Also, there are other parameters that affect protein interactions such as environmental conditions (solution, co-solvents, temperature, PH, ...) and most importantly, we do not know if these are ALL the factors.

The conventional protein engineering paradigm relies on generating mutants of what is mostly results in protein variants with similar topologies, similar molecular weights, even largely similar sequences as their parent protein. With regard to engineering topological deviations, these were mostly restricted to insertions, deletions, or circular permutations. These topological rearrangements minorly perturb the linear order of secondary structures along a sequence.

For example, we have very recently attempted to engineer mutants of the native G-CSF sequence (8-point mutants). While these mutants were expressing more than 10-fold higher compared to rhG-CSF, they were very poorly soluble and tended to aggregate. Several attempts in the literature to generate more soluble and stable mutants of rhG-CSF were not largely successful either (ref. 17-24). We argue that this inherent instability is attributable to the complex native topology of the parent structure.

In this work, we present major rearrangements on the fold, domain size, and sequence, all at once, which we justify in the introduction. We do not claim that these “controllable parameters” are the only controllable parameters that influence an active protein's half-life, folding properties, etc. We have thus rephrased the sentence as per the reviewer's suggestion as follows:

“In addition to high activity levels, additional criteria are pivotal for the successful deployment of a protein drug, such as activity half-life, folding kinetics, folding thermodynamics, solubility, and molecular weight [4, 5]. These properties depend on a set of protein-related parameters such as a protein's sequence, topology, and size.”

-line 47: The introduction stressed on improving ‘folding thermodynamics’ or ‘minimizing the loss of entropy’, yet thermodynamic parameters weren't reported. Perhaps perform DSC to report the thermodynamics of the designed variants or reword the paragraphs in the ‘Introduction’.

We have reworded the respective paragraphs accordingly.

-line 48: please explain what is meant by “topological contact order”?

We have added an extra sentence explaining the meaning of the “topological contact order” (i.e. average sequence distance of contacting residues).

-line 49-51: please clarify the phrases “simpler domain topologies” and “smaller number of folding intermediates”. We are still not able to understand structures of proteins as small as 40 residues (e.g., amyloid-beta in Alzheimer’s). The “smaller number” that authors refer to is usually not a distinct number, but an “ensemble” of intermediate structures that a protein undergoes before folding or aggregating and it is extremely difficult (if not impossible) to find all of them.

The smaller number of native contacts and the more local they are, the simpler a fold is. Certainly, the intermediate folding microstates constitute an ensemble, and we did not mean that partially folded microstates are tractably few, but in many examples (especially in large, slow-folding proteins) folding intermediates pass through spectroscopically and energetically distinct ensembles (a good example is work by Walter Englander and others). Nonetheless, what we meant was that these observable intermediate “basins” are fewer in simpler, smaller folds. We have removed the sentence on this point to avoid confusion.

-line 52-54: in general, excluding the effects of temperature and assuming that heat capacity is constant, the free energy is $dG = dH - T dS$ (G: free energy, H: enthalpy, T: temperature, S: entropy). Saying that decreasing entropy minimizes the folding free energy without specifying all the assumptions and conditions is incorrect. For example, removing long loops has definitely an effect on the number of hydrogen bonds between protein and water molecules which should be discussed, as well as the nature of amino acids in the removed loops.

Of course, a decrease in entropy yields a higher AG. We meant that a shortening of a flexible loop reduces the difference between $S_{un\ folded}$ and S_{folded} . In agreement with this idea, Zhou (Account of Chemical Research, 2004) has used polymer theory to derive the free energy cost of end-to-end restriction of a flexible loop upon folding of the rest of the protein.

To avoid the misunderstanding, we have reworded this sentence to emphasise that loop shortening reduces ΔS , and we added a theoretical reference:

“Topological simplification also tends to affect the folding thermodynamics, for instance, removing long, flexible loops tends to decrease the absolute difference in entropy between the unfolded and folded states (i.e. ΔS) [8, 9].”

Of course, we acknowledge that changes in the folding energy landscape are also subject to a complex mixture of features of the shortened loop, such as as the hydrophilicity, charges, flexibility, among others. But owing to the limited scope of the study, we only highlight the largest design manipulations done to the parent structure.

-line 76: The authors mention that of rhG-GCSF suffers from “low recombinant production yield, poor solubility and stability, and short shelf- and serum half-lives..” Please report the (values of these) parameters for G-CSF (if there are reported in literature) and the motivation of improving its thermostability. At what concentrations do these cytokines need to be stored? Moreover, the yields are increased at a cost of activity. I believe the novelty of this manuscript lies in de novo design and characterization of Boskar4 (rather than improving the properties of GCSF).

We have indeed reported (and now added a reference) on the expression yield, solubility, melting temperature, and proteolytic stability of G-CSF. G-CSF has a half-life of < 4 hours in serum, but we cannot compare this to our design in this study as we have not yet conducted pharmacokinetics experiments *in vivo*. The reviewer has a point discussing the activity:yield tradeoff for our initial designs, but now we could reach much more active ligands by enhancing the dimerisation capacity through creating design tandems.

- line 79: Please explain what has been achieved in the mentioned refs by doing classical protein engineering on rhG-CSF. How much have these studies improved the drawbacks of the pharmaceutical features so it is possible to compare it with this manuscript's results at the end.

The improvements were really minimal, at best Luo et al. (Protein Science, 2002) improved the melting temperature by 20 °C, and about 15 °C improvement was reported by Miyafusa et al (ACS Chemical Biology, 2017). Bishop et al. (Journal of Biological Chemistry, 2001) improved guanidium chloride unfolding concentration by 1.3 M. The rest of the studies did not show protein unfolding experiments, rather relying on a different shelf-life or activity duration to judge their degree of success at stabilising the protein. We think these improvements are minor as compared with our results. Therefore, we do not narrow the scope too much. Instead, the references are available for the interested readers.

-lines130-138: Since only %30 sequence identity is conserved please explain what other criteria were used in the designs and which residues were allowed to be designed in each process. Also, please explain what constraints were used on the structure during the designs.

Reordering the secondary structures across the sequence, removal of loops, and mutagenesis of the bundle itself all contributed to the loss of “alignable” similarity between the Boskars and the G-CSF template. We have now included the Rosetta “resfile” used, which designates the designable residues in the calculation. The RosettaScripts protocol, now amended to the supplement, results in only minor deviations (due to the very minor backbone moves made by the backrub protocol), hence we did not use an RMSD filter. During the molecular dynamics, we fixed all loop-distal atoms to simplify the phase space. Next, unrestrained simulations were run on a smaller number of top candidates. We indicated this in the additional supplementary methods, and the relevant materials and methods subsection.

-line130-131: the length of wild type and designed proteins are mentioned to be 174 and 123 amino acids. In Figure 1 they are shown as 163 and 119 in contact maps. Please clarify. Also, I would suggest mapping the binding site on Fig 1A.

We have updated the contact map numbering (according to PDB: 2D9Q structure), where the structured part is between residues 10 and 172. For our design, the part that was structurally modelled is 119-residue-long, so the numbering there is arbitrary. For Fig. 1A, we already map the binding site as the blue surface patch.

The authors mentioned that the refactoring process involved optimization of solvent exposed residues. Is it possible to include hydrophilic plots of G-CSF and Boskar4 or perhaps include the plots of solvent accessible surface area with respect to residue number?

As the reviewer advised, we generated hydropathy plots. We also reasoned, that it would be more useful to answer this question using estimated solvation energies of the resulting G-CSF and Boskar4 structures. Therefore, we calculated the residue-wise solvation score using the Rosetta *talaris14* scoring function. We also sought to calculate the solvation free energy using our Damietta design software (Please, see the methods section and Fig. S12). The three methods tend to show strong hydrophobic outliers in G-CSF, that are less frequent in Boskar4.

-line 178: “Most of the partial digestion of Boskar4 appears to be restricted to terminal fragments, possibly representing the unstructured hexa-Histidine and TEV cleavage purification tag.”-were these observations based on SDS-PAGE gel? Is there Mass Spec to confirm.

We have indeed carried out an MS analysis of these bands as also suggested by other reviewers. It appears that N-terminal degradation of the tag was detected, as well as a cleavage of a C-terminal 11-residue stretch (Fig. S7).

-line 327: “minimally perturbing the receptor binding epitope”: A quick ^{15}N -HSQC of Boskar4 with GCSF receptor would have helped delineate the residues involved in binding-this would have confirmed the binding epitopes/residues of Boskar 4.

We wanted to carry this out initially, but as we tried to recombinantly express the N-terminal ectodomains of the G-CSFR (both CRH and Ig-like domains, or CRH domains alone) in *E. coli* on either a pET28 or a pET32 vector (with Trx-tag), the resulting protein was of very low solubility. Our best attempts could concentrate it sufficiently enough for NMR titrations, or ITC binding studies, before it precipitated.

This is also the reason why we obtained mammalian cell-expressed G-CSFR from a commercial source (https://www.rndsystems.com/products/recombinant-human-g-csfr-cd114-protein_381-gr), and that is why we resorted to SPR for binding studies that require minute amounts of the receptor.

-line 327-8: “goal of preserving the receptor binding thermodynamics”- I’m not sure what experiments support this claim. Perhaps the authors should have performed ITC (instead of SPR) to report the thermodynamic parameters.

As mentioned above, the ITC technique requires higher concentrations and total protein amounts than SPR. Our bacterially-expressed ectodomains were of very low solubility, and in our experiments we used commercial mammalian-expressed G-CSFR, which is not supplied in quantities sufficient for ITC either.

-line 340: “here we deploy a more generalisable approach of de novo loop design that obviates the reliance on existing protein structures.”: The authors mentioned earlier that they wanted to introduce structural elements to replace loops-which implies that prior structural information was utilized in designing the new variants-pls clarify or there might be something that I’m missing.

In our strategy, we indeed relied on pre-existing structural information of the rhG-CSF, whereby we kept the structure of helical bundle itself constant, and optimized its core and surface residues of that bundle distal from the binding site. Moreover, we eliminated very long loops (one of them interrupted by helix E), replacing them by *de novo* constructed loops, after reordering the helices (which alters the bundle topology). To do this, we bridged the two pairs of N-to-C termini with 3 or 4 residue loops carrying all of the possible sequence combinations of a set of 8 or 9 amino acid alphabet. We conducted both restrained and unrestrained molecular dynamics simulations to all of these generated structures with unique loop sequences, and we chose the most conformationally stable. This was not done before in the publication describing generation of Neoleukin by Silva *et al.* (Nature, 2019), where they had to rely on grafting loops from a structure database of loops.

We are clarifying this again in the design flow chart requested by the reviewer in an earlier comment (Fig. S27). We also slightly rephrased the sentence to contrast this with the resc scaffolding strategy we have recently applied (Hernandez *et al.*, PLOS Biology, 2020).

The grouping of panels into each figure was a little hard to follow. For example, Fig 2E and 2F are discussed after Fig 4. I realize that this may be done due to other factors, but this is something to consider as it does make the article more difficult to follow.

We just kept these two panes there as the Fig. 4 became too crowded. Since we have added much more data in the revised manuscript, primarily reserving the main figures on the functional assays for our most active tandem design variants (please, see combined response to the editor and reviewers), what is now Fig. 7 is very packed and less suitable to include these two panes. The flow of the data presentation starts following the tandem designs from Fig. 5 onwards, as we moved the original functional assays figure panels into the supplement.

Please include the sequence alignment of G-CSF and all 4 variants of Boskar.

We added Table S5 to display the BLAST sequence alignments of the designs against rhG-CSF.

Throughout the manuscript, the authors mentioned that their goal is to 'rigidify' the structure, in particular, the binding epitope. Please provide an explanation as to why relaxation experiments weren't performed; otherwise, pls provide RMSD from MD simulations.

As per the reviewer's (as well as reviewer #2's) suggestion, we queried the internal dynamics using a $^{15}\text{N}\{^1\text{H}\}$ -heteronuclear NOE experiment (covering up to nanosecond-scale motions). Additionally, as per reviewer #2's suggestion, we conducted an amide proton exchange experiment (MEXICO experiment), which reports millisecond scale dynamics and is ideal for analysing helical and loop regions. The results are characteristic of a stable 4-helix-bundle (see Fig. 4D and the NMR results section).

Is there any particular reason of preferring 'median and median absolute deviation' over standard deviation or mean absolute deviation?

The median and its associated MAD are more robust to outliers than the mean, which therefore gives a better estimate of the central tendency in noisy data sets. This analysis was used for the data in Fig 2C,D, which is noisy since cells were not fixed to the plate. Although this is more physiological for the cell type used, this results in more variable measurements due to minor tilts of the plate during imaging of four regions per well. However, experiments with fixed cells (Fig. 2B) results in much lower deviations from the mean.

Table S1 suggests that the wild type is still much more active (lowest EC50). Can the authors comment on this?

The authors should discuss the pM affinity of rhG-GCSF. The rhG-GCSF is 1000-fold active than the designed variants. Include possible reasons why Boskar variants weren't as potent or didn't have high affinity to GCSF receptor.

After the more rigorous studies we did on the Boskar4 tandem constructs and by investigating the activity difference between monomeric and dimeric fractions of the Boskar4, we argue that Boskar4 is an inefficient dimeriser of G-CSFR. We think that Boskar4 displays only one binding site (*site II*), and is minorly dimeric, hence having lower activity (albeit nanomolar) than rhG-CSF.

Boskar4_st2, which is a short-linker tandem of the Boskar4 binding domain, has a comparable affinity to G-CSFR as G-CSF, i.e. $K_d(\text{Boskar4_st2}) \approx 5 \text{ nM}$, and $K_d(\text{rhG-CSF}) \approx 1 \text{ nM}$. The same was observed in the NFS-60 proliferation assays, where $EC_{50}(\text{Boskar4_st2}) \approx 8 \text{ pM}$, and $EC_{50}(\text{rhG-CSF}) \approx 2 \text{ pM}$.

Also the spacing of the dimerising domain seemed to influence the activity: Although both Boskar4_t2 and Boskar4_st2 show almost the same affinity to the receptor, Boskar4_st2 is more than 10-fold more active than Boskar4_t2. While Boskar4_st2 is already a picomolar activator, we still think there is more activity to be gained if two (or possibly even more) Boskar4 domains are constructed in tandem with rigid linkers that impose the ideal orientation needed for maximal receptor activation. We already started working on investigating this for a future study.

Pls mention here that the rhG-GCSF is also referred as Filgrastim. This is stated in Figure legend or Methods.

Yes, it is. We also changed in the figures to "rhG-CSF".

Circular dichroism: is it possible that the lower Tm (of rhG-GCSF) is due to breakage of disulfide bond? It is possible that the absence of disulfide bond may improve thermostability and

reversibility of the designed variants. What if a reducing agent is added to the buffer-can it improve the thermostability/irreversibility of the protein?

To test this hypothesis, we have conducted melting experiments of rhG-CSF in three conditions: 1) in a mixture of reduced and oxidised glutathione (to allow for reversible formation of disulfide bridges), 2) in dithiothreitol (to keep all cysteines reduced), and 3) in PBS buffer only.

As recording CD spectra was not possible due to the high UV absorbance of these reagents, we measured the melting curves using nanoscale-DSF.

These results showed that the behavior of G-CSF is very similar with or without GSH/GSSG. However, in a reducing environment (with DTT), G-CSF is substantially less stable. In all cases the unfolding was irreversible, and in the first two cases, the nanoDSF results matched that of CD. So we can rule out that the disulfide bond breakage is the reason of the instability of the irreversibility of the unfolding (Fig. S3-S5).

Pls report the solvent and protein concentration used for NMR.

Done.

Is there evidence that G-CSF of human origin would be active in zebrafish?

Our data is consistent with previous observations (Gianoncelli *et al.* Journal of Chemistry, 2019). The use of Filgrastim (rhG-CSF) for neutrophil induction in zebrafish embryos has been established in two different labs participating in this study.

I would suggest adding Figure 2F to figure 5.

Now Fig. 7 (functional assays figure; previously Fig.5) is overcrowded, and is focused on the design tandems. Why we chose to keep Fig. 2F where it is, in the part of the manuscript where we describe the activity of the original designs.

Minor Revisions (typo's, clarifications, etc.):

- Lines 47-57: While the terminology being used here may be known to people in the field, it may be difficult to follow for the broader scientific community. Consider defining terms such as “topological contact order” or “downhill folding”, or describing why these are significant in layman’s terms.

To make our work more accessible to more readers, we have clarified “topological contact order” and removed the sentence containing “downhill folding”.

- Line 92: Need a reference for this.

Added.

- Line 111: Spell out the name for “MD” before using abbreviations.

Done.

- Line 130: Edit “were 123 amino acid-long...” to “were 123 amino acids long...”

Corrected.

- Line 136: Spell out the name for “PTM” before using abbreviations.

Done (Line 100).

- Line 137: "...proteins were designed not contain..." to "...proteins were designed not to contain..."

Corrected.

- Table S1: The "50" in "EC50" in the fifth column should be

subscripted. Corrected.

- Supplementary Movies: What were the doses used when treating these cells for the movies? We added the concentrations in the legend.

- Line 163: There is a space after ">" in "> 20 mg/ml". Earlier instance in Line 161 has no space. Either is fine, but be consistent.

Done.

- -line 175: "NE enzymatically degrades G-CSF activity": pls remove the word

'activity'. Done.

- Line 254: change "a concentration of 50 nM for designs" to "...for each design" This section was rewritten to indicate also the concentrations of the new tandem designs.

- Line 288, pls spell out 'dpf'.

Done.

- Line 322: "...receptors ectodomains..." to "receptor's ectodomains"

We changed it to "receptors' ectodomains", as even one cytokine can dimerise two different receptors (please, see Sprangler et al., Annual Reviewers in Immunology, 2014).

- Line 402: with the computational modeling section, the proteins were modeled in isolation and were significantly less functional than the naturally occurring proteins. Would it be possible to model them binding with the intended receptor to simultaneously optimize both the structure and the binding?

That indeed would be the ideal modelling environment, however, we aimed to avoid this for different reasons:

- The binding interface of G-CSF:G-CSFR (site II) is highly hydrophilic with trapped waters, a situation where classical force fields (let alone the existing protein design scoring functions) are not very accurate in modelling.
- Including the receptor ectodomains to either the design calculations would add the load of sampling combinatorial rotamers of the receptor sequence that is combined with sampling rotamers and mutations on the ligand side. Given a constant computing footprint, this reduces the overall mutants and rotamers sampled to stabilise the ligand structures.
- The same applies to the molecular dynamics simulations, where the added atoms would not just need to be run for shorter time spans, but also the presence of the receptor atoms

would create a bigger system that needs longer simulation time (due to the larger phase space).

As we describe in the manuscript, the structural deviations between bound and free G-CSF are minimal. Moreover, the interface residues were not mutated, so the extra layer of receptor atoms around the interface are even farther from the actual mutable sites in the ligand.

- Line 405: “The reduced bundle was determined to be the minimal helical bundle sufficient...” How was this determined? Is this stated in literature (which would require a citation) or determined by your lab (which should have some discussion on how this was determined).

We changed this to “the minimal 4-helix-bundle”, as the up-down 4-helix-bundle is the minimal topology.

- Line 459: “cOmpete” to “complete”

That was not a typo; the company calls it “cOmpete”:
<https://www.sigmaaldrich.com/DE/en/product/roche/coro>.

- Figure S6 legend-typo ‘monoric’-pls change it to ‘monomeric’

Corrected.

Reviewers' Comments:

Reviewer #1:

Remarks to the Author:

The authors have satisfactorily answered to my comments and I recommend the publication of this manuscript.

Reviewer #2:

Remarks to the Author:

In their revised manuscript entitled "A topological refactoring design strategy yields highly stable granulopoietic proteins", Skokowa et al. have provided important additional data that strengthened their study (in particular the functional aspects of the paper) and improved the overall quality of the manuscript. The authors have clarified most of my initial concerns and I am happy about the changes, but I would still like to comment on these specific points:

- The authors misunderstood me if they thought my concerns were centered on the CoMAND method of determining the Boskar4 solution structure. Rather, my concerns came from the fact I was not provided with any mean to evaluate the structure they have presented. I would have raised similar concerns even if more traditional structural calculation methods were used but with no sufficient data presented. That said, I personally think the CoMAND method relies too much on assumptions and (potentially risky) predictions, but I do acknowledge it can perform well at least for small and simple-folded proteins like Boskar4 (because they are very likely to yield largely-resolved and easy to analyze 3D-CNH-NOESY spectra). I appreciated the efforts of the authors in extending their structural calculation protocol also to include some important core NOEs they have manually assigned and inspected.

- I was happy to see a 2D 1H-15N HSQC spectrum of Boskar4 and the TALOS analysis from experimental chemical shifts which gave support to the presented structure. However, I would have preferred the most important NOESY strips to be shown in a figure (as I asked) rather than just provided as the NOE list of Supplementary Table 3. I have no reason to doubt the NOE assignment made by the authors, but again this leave the reviewers in the impossibility to verify that no obvious mistakes have been done. The authors stated such a figure is included in the PDB/BRMB submission, but the entries are locked (hold for publication) and no updated PDB report was submitted with the rebuttal.

- While I understand that the authors are not inclined to measure a 2D 1H-15N HSQC spectrum of Boskar4 before and after the temperature ramp to assess its stability (complementary to the CD data), I do not understand their argument for rejecting the experiment on the basis that NMR samples were stable at 40°C over several days. In fact, data shown in Figure 3 reveals that even G-CSF is stable up to 57°C. Rather, I think this experiment could be useful (and potentially interesting for the project) for assessing whether the high temperature (>90°C) and subsequent slow cool-down can affect Boskar4 monomer-dimer equilibrium and shift the population toward one uniform species.

Additionally, in light of the new data and protein constructs (Boskar4_t2 and Boskar4_st2), I think it has now become very important to fully discuss the protein oligomeric issue even if in a semi-speculative way. I am well aware that additional thorough studies are probably required to fully address this matter, but I think that the many data collected by the authors already support well their hypothesis that Boskar4 is not able to efficiently dimerize G-CSFR as G-CSF does. This is probably due to the fact that residues playing this role have been removed when designing the engineered protein. Funny to see that the residual dimerization propensity (or a design artefact) provided to the authors the important initial clue that led to Boskar4_t2 and Boskar4_st2 design!

- First of all, I do not think it makes much sense to present SPR binding data on inhomogeneous samples (and in fact in the originally submitted manuscript I thought only the monomeric pool was used). Therefore, I suggest to include in the manuscript the data of isolated monomeric and dimeric

forms of Boskar4 (that should be already available, as mentioned by the authors). I think this would be very important to clearly show the trend of improvement passing from monomeric to dimeric species: monomeric Boskar4 -> dimeric Boskar4 -> Boskar4_t2 -> Boskar4_st2.

- Second, I would inspect again the NOESY spectra already available for Boskar4 and search for intermolecular NOE peaks of the dimeric form. If the complex is not symmetric they may be readily spotted (depending on spectral overlap). What I think would be useful is to understand whether the dimeric unlinked species has the same tail-to-head arrangement of Boskar4_t2 and Boskar4_st2.

- Finally, and most important of all, I recommend collecting additional SPR binding data on monomeric Boskar3, dimeric Boskar3 and Boskar3_st2. Mixed constructs (e.g., Boskar4-GGGGSS-Boskar3) would be useful too. The point here is to see whether the measured affinities will fit in the expected trend. If so, I think this will be a compelling experiment in support of the authors' hypothesis and will provide the missing rationale for the design of the linked tandem proteins.

Minor revisions:

1. Figure 4, panel D: I think the figure would benefit from drawing the secondary structure elements of the protein on top of the exchange rates and NOE data. This will make easier to relate the new data to the protein structure.

2. Supplementary Fig. 7: protease, Boskar 4, G-CSF and marker ladder bands should be labeled. Although in the Materials and Methods section it is stated that identical protein concentrations were used, protease bands in the Boskar 4 gels are much fainter compared to their counterparts in the G-CSF gels. This could be explained assuming that Boskar 4 gels have been loaded less (it is possible since Boskar 4 bands are fainter too); still, in the case of Cathepsin G it seems the protease/protein ratio is different in the two experiments. Please comment on this regard.

3. Supplementary Fig. 8: A77 is obviously folded in the spectrum. Please indicate it in the figure or in the related caption.

4. Caption of supplementary Fig. 9: there is a typo in line 1373 (CoAMND); both CoMAND and TALOS angles are said to be plotted as green bars, but the latter is obviously blue.

Reviewer #3:

Remarks to the Author:

While the authors attempted to address the issues expressed during the first submission, they did not explicitly specify which figures had been updated and which modifications had been made (both in the text and figures), making it difficult to notice the improvement. For instance, the authors' response to one of the queries:

Figure 5D, It is unclear where the CD45+CD16+CD66b+ data were derived from? Based on the Supplemental Figure 11, there was no CD66b staining.

We corrected supplemental figure 21.

Additionally, it is difficult to imagine how the CD45+CD11b+CD15+ and CD45+CD15+CD16+ populations be so similar when quantified as absolute counts when their frequency is quite different. I suggest the authors to re-check the data.

The data are correct. The total number of cells at day 14 of differentiation was different in each well/stimulation condition. We calculated absolute cell numbers based on the total number of cells. Additionally, suppl. Figure 21 is showing representative FACS blots of one well per stimulation condition, and the percentage of respective cell populations in the second well may slightly vary.

This is a little confusing as the author replotted their FACS analysis, these plots are different from the original plots (original Suppl Figure 11->Suppl Figure 21), but the absolute counts (original Figure 5D -> Suppl Figure 22) remained the same.

More importantly, all the experiments, including new data in the revised manuscript such as Figures

7C and 8D, demonstrated that recombinant GSCF is still more potent in their functional activities. This begs the question, what is the advantage of using topological refactoring to design novel therapeutic proteins?

Reviewer #4:

Remarks to the Author:

The authors gave adequately addressed the issues raised.

REVIEWER COMMENTS

Reviewer #1 (Remarks to the Author):

The authors have satisfactorily answered to my comments and I recommend the publication of this manuscript.

We are thankful to the reviewer's contribution and time to improve our manuscript.

Reviewer #2 (Remarks to the Author):

In their revised manuscript entitled "A topological refactoring design strategy yields highly stable granulopoietic proteins", Skokowa et al. have provided important additional data that strengthened their study (in particular the functional aspects of the paper) and improved the overall quality of the manuscript. The authors have clarified most of my initial concerns and I am happy about the changes, but I would still like to comment on these specific points:

- The authors misunderstood me if they thought my concerns were centered on the CoMAND method of determining the Boskar4 solution structure. Rather, my concerns came from the fact I was not provided with any mean to evaluate the structure they have presented. I would have raised similar concerns even if more traditional structural calculation methods were used but with no sufficient data presented. That said, I personally think the CoMAND method relies too much on assumptions and (potentially risky) predictions, but I do acknowledge it can perform well at least for small and simple-folded proteins like Boskar4 (because they are very likely to yield largely-resolved and easy to analyze 3D-CNH-NOESY spectra).

We appreciate the reviewer's concerns on the deposition process for CoMAND structures. Many of these have now been resolved, as CoMAND is now a recognised method within the PDB deposition framework. Also please note that the full PDB submission has now been released (PDB: 7NY0, BMRB: 34613). Part of our motivation in developing CoMAND has been to provide a robust method of validation of NMR structures against the input data, and we hope the deposition process will continue to improve, helping achieve this.

I appreciated the efforts of the authors in extending their structural calculation protocol also to include some important core NOEs they have manually assigned and inspected.

These restraints were certainly useful in validating the previous ensemble and we plan to include such restraints in future versions of the protocol.

- I was happy to see a 2D 1H-15N HSQC spectrum of Boskar4 and the TALOS analysis from experimental chemical shifts which gave support to the presented structure. However, I would have preferred the most important NOESY strips to be shown in a figure (as I asked) rather than just provided as the NOE list of Supplementary Table 3. I have no reason to doubt the NOE assignment made by the authors, but again this leave the reviewers in the impossibility to verify that no obvious mistakes have been done. The authors stated such a figure is included in the PDB/BRMB submission, but the entries are locked (hold for publication) and no updated PDB report was submitted with the rebuttal.

We had included example strips for residue A97 in (now Supplementary Figure 11), which we thought adequately illustrated the quality of the data and the model selection process. Supplementary Figure 12 also added detail of the backbone polymorphism at residue E23 in the end C-terminal cap of the first helix. We have now added an extra figure with some assigned strips as Supplementary Figure 10. We had mentioned that all such strips are included in the PDB submission (as raw data, rather than a figure) and this has now been released. We are also attaching the PDB structure validation report in this revision (and it is also available at the PDB itself).

- While I understand that the authors are not inclined to measure a 2D 1H-15N HSQC spectrum of Boskar4 before and after the temperature ramp to assess its stability (complementary to the CD data), I do not understand their argument for rejecting the experiment on the basis that NMR samples were stable at 40°C over several days. In fact, data shown in Figure 3 reveals that even G-CSF is stable up to 57°C. Rather, I think this experiment could be useful (and potentially interesting for the project) for assessing whether the high temperature (>90°C) and subsequent slow cool-down can affect Boskar4 monomer-dimer equilibrium and shift the population toward one uniform species.

As per the reviewer's request, we have prepared new 15N-labelled samples and collected 2D HSQC spectra before and after a temperature ramp till 100°C and back to 25°C at 1°C/min (similar to the CD experiment). The result of the experiment has been included as Supplementary Figure 13. The 1H-15N HSQC spectrum completely recovers after heat treatment.

Additionally, in light of the new data and protein constructs (Boskar4_t2 and Boskar4_st2), I think it has now become very important to fully discuss the protein oligomeric issue even if in a semi-speculative way. I am well aware that additional thorough studies are probably required to fully address this matter, but I think that the many data collected by the authors already support well their hypothesis that Boskar4 is not able to efficiently dimerize G-CSFR as G-CSF does. This is probably due to the fact that residues playing this role have been removed when designing the engineered protein. Funny to see that the residual dimerization propensity (or a design artefact) provided to the authors the important initial clue that led to Boskar4_t2 and Boskar4_st2 design!

- First of all, I do not think it makes much sense to present SPR binding data on inhomogeneous samples (and in fact in the originally submitted manuscript I thought only the monomeric pool was used). Therefore, I suggest to include in the manuscript the data of isolated monomeric and dimeric forms of Boskar4 (that should be already available, as mentioned by the authors). I think this would be very important to clearly show the trend of improvement passing from monomeric to dimeric species: monomeric Boskar4 -> dimeric Boskar4 -> Boskar4_t2 -> Boskar4_st2.

The reviewer's conclusion is correct. For more clarity we have conducted full SPR titrations for the dimeric and monomeric species of both Boskar3 and Boskar4 and included them in Supplementary Figure 20. The tandem constructs B3_st2, B4_st2, and the mixed domain tandem B3_B4 (the reviewer suggested) display tighter apparent binding than the isolated dimeric species, which in turn bind tighter than the monomeric species.

- Second, I would inspect again the NOESY spectra already available for Boskar4 and search for intermolecular NOE peaks of the dimeric form. If the complex is not symmetric they may be readily spotted (depending on spectral overlap). What I think would be useful is to understand whether the

dimeric unlinked species has the same tail-to-head arrangement of Boskar4_t2 and Boskar4_st2.

The spectra in general show no signs of multiple conformations that would be consistent with a minor population of dimer. Any monomer-dimer equilibrium must therefore take place on timescales fast enough to average chemical shifts. However, any exchange of this type would be expected to result in concentration-dependent chemical shift changes. We therefore performed a dilution series, observing that the spectrum is unaltered upon four-fold dilution (Supplementary Figure 13). We conclude that the sample is overwhelmingly monomeric under the measurement conditions. All NOE cross peaks we observed in the spectrum are well explained within the monomer structure.

- Finally, and most important of all, I recommend collecting additional SPR binding data on monomeric Boskar3, dimeric Boskar3 and Boskar3_st2. Mixed constructs (e.g., Boskar4-GGGGSS-Boskar3) would be useful too. The point here is to see whether the measured affinities will fit in the expected trend. If so, I think this will be a compelling experiment in support of the authors' hypothesis and will provide the missing rationale for the design of the linked tandem proteins.

As the reviewer suggested we tested the st2-variant of the "mixed" construct (i.e. Boskar3-GGGGSS-Boskar4; we refer to it as Boskar3_Boskar4 in the manuscript). It showed an affinity of 2.6 nM, which is very similar to that of B4_st2 that binds with a K_D value of about 6 nM. We included these additional results in Table 1, and Supplementary Figure 20. We also assayed the proliferative EC50 for the monomeric and dimeric species of Boskar3, as well as the Boskar3_st2 and the Boskar3_Boskar4 tandems. The dimeric and tandem forms of the designs clearly showed higher proliferative activity than their monomeric counterparts, as expected (Supplementary Figure 21).

Minor revisions:

1. *Figure 4, panel D: I think the figure would benefit from drawing the secondary structure elements of the protein on top of the exchange rates and NOE data. This will make easier to relate the new data to the protein structure.*

We have modified the figure, thanks for the suggestion.

2. *Supplementary Fig. 7: protease, Boskar 4, G-CSF and marker ladder bands should be labeled.*

Although in the Materials and Methods section it is stated that identical protein concentrations were used, protease bands in the Boskar 4 gels are much fainter compared to their counterparts in the G-CSF gels. This could be explained assuming that Boskar 4 gels have been loaded less (it is possible since Boskar 4 bands are fainter too); still, in the case of Cathepsin G it seems the protease/protein ratio is different in the two experiments. Please comment on this regard.

We assure the reviewer that the loaded concentrations were matched. However, Boskar constructs Coomassie staining is mostly weaker, perhaps due to its lower molecular weight, or incomplete denaturation. We have anyway repeated these assays (new Supplementary Figure 7), and our results are showing the same picture with some partial degradation observed for G-CSF upon cathepsin G treatment. We have updated the methods and results sections accordingly.

3. *Supplementary Fig. 8: A77 is obviously folded in the spectrum. Please indicate it in the figure or in the related caption.*

We indicated it in the caption, also for the folded A121.

4. *Caption of supplementary Fig. 9: there is a typo in line 1373 (CoAMND); both CoMAND and TALOS angles are said to be plotted as green bars, but the latter is obviously blue.*

We have corrected the mistake in the caption.

Reviewer #3 (Remarks to the Author):

While the authors attempted to address the issues expressed during the first submission, they did not explicitly specify which figures had been updated and which modifications had been made (both in the text and figures), making it difficult to notice the improvement. For instance, the authors' response to one of the queries:

Figure 5D, It is unclear where the CD45+CD16+CD66b+ data were derived from? Based on the Supplemental Figure 11, there was no CD66b staining.

We corrected supplemental figure 21.

We apologise for this confusion, but as the document size grew, we had to move several items to the supplement. The representative FACS histograms (initial Suppl. Figure 11, Suppl. Figure 25 now) of the experiments initially presented in Figure 5D (Suppl. Figure 22 now) are correct in the revised version. These are representative diagrams of one out of three independent experiments performed in technical replicates. In the previous revised version and also here, we decided to select more representative FACS images while replacing the initially submitted ones.

Additionally, it is difficult to imagine how the CD45+CD11b+CD15+ and CD45+CD15+CD16+ populations be so similar when quantified as absolute counts when their frequency is quite different. I suggest the authors to re-check the data.

The data are correct. The total number of cells at day 14 of differentiation was different in each well/stimulation condition. We calculated absolute cell numbers based on the total number of cells. Additionally, suppl. Figure 21 is showing representative FACS blots of one well per stimulation condition, and the percentage of respective cell populations in the second well may slightly vary.

This is a little confusing as the author replotted their FACS analysis, these plots are different from the original plots (original Suppl Figure 11->Suppl Figure 21), but the absolute counts (original Figure 5D -> Suppl Figure 22) remained the same.

Please, see our comment above.

More importantly, all the experiments, including new data in the revised manuscript such as Figures

7C and 8D, demonstrated that recombinant G-CSF is still more potent in their functional activities. This begs the question, what is the advantage of using topological refactoring to design novel therapeutic proteins?

The constructs presented in this manuscript are much more stable, conformationally homogeneous, and easier and cheaper to produce compared to rG-CSF. These biophysical properties are prerequisites for high bioavailability and serum half-life, making the designs promising initial candidates for G-CSF replacement therapies. To evaluate these assumptions, regarding the proteolytic stability of the constructs, we have already started to investigate their oral bioavailability in zebrafish, with promising initial results. We are currently planning to investigate their bioavailability with and without fusion to transferrin receptor binders (<https://doi.org/10.1073/pnas.2021569118>). This is of particular interest as orally-administrable rhG-CSF could be great value for instance for congenital neutropenia patients who are chronically treated with G-CSF by daily s. c. administrations life-long. Particularly as unmodified rhG-CSF is not bioavailable orally (<https://doi.org/10.1023/A:1016089503186>).

Beyond the scope of this study, we have also identified mutants of our designs with improved affinity to the G-CSFR, and we are also exploiting the idealised helical structure of the designs to create rigid fusions with different receptor dimerization geometries to identify more efficient receptor signaling configurations. Conversely, we also started design variants of the Boskar designs that are homogeneously monomeric, and can act as inhibitors of G-CSFR signaling.

We are convinced that the important piece of information presented in this manuscript is of great importance giving the readers essential insights on the prospects of protein (cytokine) design for medical and also biological needs. Exemplary, in our Severe Chronic Neutropenia International Registry (SCNIR), we have identified several cases of congenital neutropenia patients where the Ig-like domain of the G-CSFR is mutated by residues that are modelled to destroy the G-CSF-binding interface at site III. Such patients are poorly responding to rhG-CSF treatment, and given the Ig-like domain-independent activity of the Boskar designs, these patients may respond better to treatment using our designs. We are currently testing the activity of our designs against these genotypic aberrations in G-CSFR. In the same vein, neutropenia patients with hyperactive ELANE mutants may also benefit from the proteolytic resistance of our designs.

Reviewer #4 (Remarks to the Author):

The authors gave adequately addressed the issues raised.

We are again grateful to the reviewer's time and useful input.

Reviewers' Comments:

Reviewer #2:

Remarks to the Author:

In their revised manuscript entitled "A topological refactoring design strategy yields highly stable granulopoietic proteins", Skokowa et al. have adequately addressed all my comments and issues raised. I believe the manuscript has been greatly improved since the original submission and I congratulate the authors for their hard work.

Minor revisions:

Several supplementary figures are wrongly referenced in the manuscript. Please double check. I found obvious mistakes in lines 338, 348, 350, 352, 360, 362, 400, 545, 750 and 1487

Reviewer #3:

Remarks to the Author:

While the authors attempted to address concerns stated in the revision, they failed to highlight the modifications made in this revision, making it extremely difficult to see the improvement. Furthermore, the authors should discuss in their text (discussion section) the advantage of utilizing topological refactoring to create novel therapeutic proteins even though rhGSCF is still far more potent than their constructs.

REVIEWERS' COMMENTS

Reviewer #2 (Remarks to the Author):

In their revised manuscript entitled "A topological refactoring design strategy yields highly stable granulopoietic proteins", Skokowa et al. have adequately addressed all my comments and issues raised. I believe the manuscript has been greatly improved since the original submission and I congratulate the authors for their hard work.

Minor revisions:

Several supplementary figures are wrongly referenced in the manuscript. Please double check. I found obvious mistakes in lines 338, 348, 350, 352, 360, 362, 400, 545, 750 and 1487

Many thanks for the reviewer's input and advice. We have gone through these lines and corrected the references to the figures.

Reviewer #3 (Remarks to the Author):

While the authors attempted to address concerns stated in the revision, they failed to highlight the modifications made in this revision, making it extremely difficult to see the improvement. Furthermore, the authors should discuss in their text (discussion section) the advantage of utilizing topological refactoring to create novel therapeutic proteins even though rhGSCF is still far more potent than their constructs.

We would like to thank the reviewer for his input, drawing his attention to the fact that rhG-CSF is only 5-fold more active on a molar scale. We have accordingly wrote an additional paragraph discussing some foreseen therapeutic applications and advantages of the Boskar designs, which we are currently developing.